# MULTI-AGENT IMITATES ENTERPRISE DYNAMICS

## ABSTRACT

Imitating enterprise dynamics characterized by volatility, long-horizon coordination, and decision-making offers executives and operational teams a structural understanding of the organization at a much lower cost. Existing LLM-based agent systems have great potential to simulate these activities, yet they still face three challenges to understand the dynamics at the enterprise scale in terms of *structure*, *strategy*, and *operation*. This motivates us to propose *TaskWeave*, a novel LLM-based multi-agent framework that aims to imitate complex enterprise dynamics. Inspired by theories in the fields of control and business management, our *TaskWeave* operates at three levels: strategic (executives, e.g. generating phase plans), tactical (coordination, e.g. scheduling resource allocation), and operational (task execution, e.g. leveraging multiple tools), simulating modern enterprise dynamics in an end-to-end manner. Our *TaskWeave* instantiates an IT company to simulate year-long operations, demonstrating diverse enterprise dynamics. Experiments, including human evaluations, show that it improves performance with less real-world overhead, while also generating internal data as a downstream task and enabling interactions with external contexts.

## 1 INTRODUCTION

Modern enterprises operate as highly dynamic ecosystems (Teece, 2007; Zollo et al., 2016) in unprecedented volatility, where strategic decisions must account for rapidly evolving markets, internal stakeholder dynamics, and socioeconomic uncertainties. Accurate imitation of these dynamics offers a structured understanding of organizational behavior with *lower real-world overhead*. Previous studies can be categorized into three groups, including equation-based ones (Zimmer, 2010) that utilize physical laws to model system behaviors, data-driven ones (Ghadami & Epureanu, 2022) that leverage machine learning to approximate complex systems, and agent-based ones (Rolón & Martínez, 2012) that simulate systems as interactions between autonomous agents. Early agent-based methods (Drogoul et al., 2002) rely on mathematical paradigms such as Kermack-McKendrick (Macal, 2010) and discrete event models (Borshchev & Filippov, 2004). Recently, numerous LLM-based agentic systems, which involve single or cooperative multi-agents (Park et al., 2023), have been widely used in various domains, such as planning (Chen et al., 2024b), task decomposition (Huang et al., 2024), and scenario simulation (Li et al., 2024b). These advances show great potential to simulate system dynamics, such as economics (Li et al., 2024c; Yang et al., 2025), tiny society (Xu et al., 2023; Gao et al., 2023b), social media (Yao et al., 2024; Yukhymenko et al., 2024).

Although effective, existing LLM-based agentic systems, such as AutoGen (Wu et al., 2023), BabyAGI (Nakajima, 2023) and CAMEL (Li et al., 2023) struggle to produce accurate simulations of enterprise dynamics primarily due to three challenges: ❶ *Structure dynamics*: Enterprise structures (Galbraith, 2014) vary widely across sectors, scales, and workflows, thus requiring agents to be capable of modeling diverse coordination patterns while remaining structurally grounded. ❷ *Strategy dynamics*: Enterprise processes (Xu et al., 2008)are intrinsically hierarchical and temporally layered, and this entails agents to yield planning strategies that ensure coherence across long-horizon goals and fine-grained execution (Hou et al., 2024). ❸ *Operation dynamics*: Organizational workflows (Chebbi et al., 2006) are context-dependent and task-driven, where each action builds on evolving plans, prior progress, and external socioeconomic uncertainties. Hence, it requires agents to efficiently fuse heterogeneous clues while maintaining semantic continuity across tasks.

To address these challenges, this paper proposes *TaskWeave*, an LLM-based multi-agent framework that aims to imitate complex enterprise dynamics (show in Figure 1) through hierarchical planning,

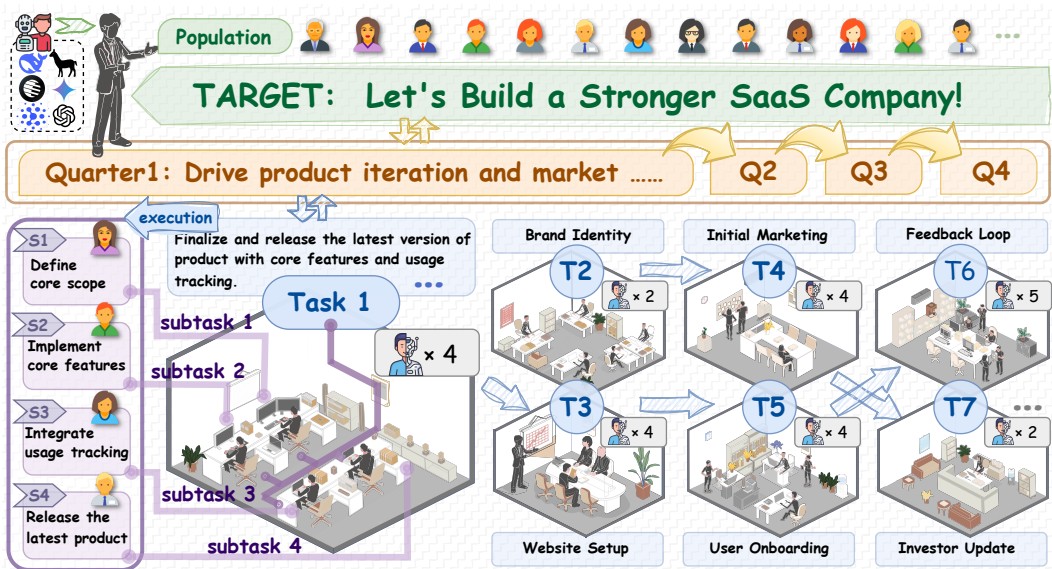

Figure 1: An overview of enterprise dynamics, derived from corporate strategy, generate interdependent tasks that are decomposed into subtasks and assigned to LLM-instantiated agents.

role-based collaboration, and context-aware execution. Inspired by recursive control theory (Deming, 1986), strategic management (Anand & Ward, 2002; Mintzberg, 1979), and division-of-labor theory (Becker & Murphy, 1992), our *TaskWeave* operates at three levels: **strategic** (executive decision-making), **tactical** (departmental and individual coordination), and **operational** (task execution), simulating modern enterprise dynamics in an end-to-end manner.

We first instantiate tens and hundreds of diverse LLM-based agents with structured delegation reflecting real-world sectors and roles, and workers, and then assign them to execute interdependent tasks under temporally evolving goals. *At the strategic level*, we present a novel control cycle termed as Formulate-Partition-Diagnose-Align (FPDA) to simulate long-term business activities with certain objectives. *At the tactical level*, agents dynamically shift between orchestrator and executor roles, enabling structured delegation, plan refinement, and inter-agent alignment across organizational tiers. *At the operational level*, agents enhance execution context via structured memory, dependency tracking, and external tool access, enabling tool-augmented reasoning with continuity and traceability in dynamic workflows. We conduct a comprehensive evaluation of TaskWeave across multiple core dimensions. Experimental results show that the proposed framework produces diverse task trajectories, maintains long-horizon strategic coherence, and generates enterprise data with richer contextual and semantic content compared to baseline systems, while also enabling interactions with external contexts and demonstrating generalizability across different backgrounds.

Our **key contributions and empirical insights** are summarized as follows.

- We propose *TaskWeave*, a novel framework that simulates enterprise dynamics through structured multi-agent collaboration. By instantiating LLM-based agents with stratified roles and coordinating them through hierarchical planning and execution, TaskWeave captures the temporal, structural, and contextual complexity.

- We introduce a three-level orchestration, including strategic planning, tactical delegation, and operational activities. It empowers agents to align global intent while executing fine-grained actions, effectively mirroring enterprise coordination. By doing so, we can properly imitate the three challenging enterprise dynamics, providing insights for executives and operational teams.

- We conduct comprehensive empirical studies across various dimensions, demonstrating TaskWeave's effectiveness in imitating coherent and diverse enterprise behavior. Experiments further indicate its potential for scalable organizational simulation, with generalizability to diverse scenarios. *We show that TaskWeave can assist management teams and academics in driving performance in rapidly evolving ecosystems with less real-world overhead, while generating enterprise data that supports downstream analysis and decision-making.*

## 2 RELATED WORK

**Enterprise-Level Process Simulation.** Corporate activity has long been modeled through *Business Process Management (BPM)* pipelines that specify task flows. Early work emphasized rule-driven formalisms such as BPMN (van der Aalst, 2011; Dumas et al., 2018), later extended with re-design (Reijers, 2003), declarative models (Pesic et al., 2007), and process mining from execution logs (van der Aalst, 2011; 2013). Even modern BPM engines rely on *static* templates and pre-defined logic, limiting their ability to capture the dynamic and adaptive nature of real enterprise coordination.

**Multi-Agent Systems (MAS):** MAS (Hong et al., 2024; Park et al., 2023) mark a paradigm shift by integrating LLMs' reasoning, planning (Hu et al., 2024), and communication (Gao et al., 2024; Li et al., 2024a) into agent-based frameworks. Leveraging pre-trained LLMs, these systems exhibit emergent planning skills, natural language collaboration, and contextual memory, enabling complex task decomposition, scenario simulation (Yang et al., 2025), and adaptive behavior modeling. Recent surveys (Chen et al., 2025) classify their trajectories into three: (i) collaborative task-solving via role-aligned agents (Islam et al., 2024; Shen et al., 2024; Yu et al., 2024; Du et al., 2023), (ii) realistic simulation of social (Mou et al., 2024; Li et al., 2024b; Gao et al., 2023c; Park et al., 2022; Wang et al., 2024; Pan et al., 2024; Xu et al., 2024; Gong et al.; Li et al., 2024c) and physical dynamics (Gao et al., 2023a; Zou et al., 2023), and (iii) meta-evaluation and training of generative agents through structured multi-agent interactions (Chen et al., 2024a; Zhao et al., 2024; Shi et al., 2024; Gao et al., 2025; Liu et al., 2024a). Meanwhile, most existing frameworks remain constrained by a focus on short-horizon planned actions, with limited coordination and temporal continuity, which restricts their applicability to realistic enterprise simulations.(detailed comparison in Appendix D)

## 3 METHOD

### 3.1 FRAMEWORK OVERVIEW

As shown in Figure 2, TaskWeave simulates complex enterprise behavior as structured populations of LLM-based agents, coordinated through a comprehensive three-level orchestration: **strategic**, **tactical**, and **operational**. This end-to-end simulation is supported by three core functional modules:

**Structured Agent Population.** Agents are instantiated from enterprise metadata and organized into a graph that mirrors real-world relationship, forming the structural backbone for coordinated planning.

**Hierarchical Task Propagation (Strategic–Tactical).** At the strategic level, the system perceives both internal and external conditions and refines global intent via the Formulate–Partition–Diagnose–Align (FPDA) cycle; at the tactical layer, agents alternate between orchestrator and executor roles to decompose goals, delegate vertically, and align horizontally.

**Context-Aware Task Execution (Operational).** At the operational layer, agents operate with structured memory and tool access, retrieving historical outputs and resolving task dependencies. This supports context-grounded reasoning and traceable output aligned with evolving objectives.

### 3.2 STRUCTURED AGENT POPULATION

To capture organizational role dynamics, we construct a Structured agent population. Each agent is initialized with identity descriptors and functional roles, and embedded within a fixed delegation topology that encodes permissible coordination flows.Given organizational metadata $\mathbb{B}$ (e.g., industry, scale, departmental layout), we instantiate $N$ agents $\mathcal{A} = \{a_i\}_{i=1}^N$, each defined as:

$$a_i = (\phi_i, \rho_i), \quad \phi_i \sim \mathcal{P}_\phi, \quad \rho_i \sim \mathcal{P}_\rho(\mathbb{B}) \tag{1}$$

where $\phi_i$ captures immutable identity traits (e.g., tenure, background), and $\rho_i$ denotes role descriptors (e.g., title, responsibility), sampled from structured distributions conditioned on metadata $\mathbb{B}$.

To model hierarchical delegation, we partition the population into $T$ tiers:

$$\mathcal{A}^{[t]} = \{a_i \in \mathcal{A} \mid \rho_i \in \mathcal{R}^{[t]}\}, \quad \bigcup_{t=1}^T \mathcal{A}^{[t]} = \mathcal{A}, \quad \mathcal{A}^{[t]} \cap \mathcal{A}^{[t']} = \emptyset \text{ if } t \neq t' \tag{2}$$

Here, $\mathcal{R}^{[t]}$ defines the role family assigned to tier $t$, derived from $\mathbb{B}$ to reflect enterprise stratification. Furthermore, to enable recursive planning across strategic, tactical, and operational layers under

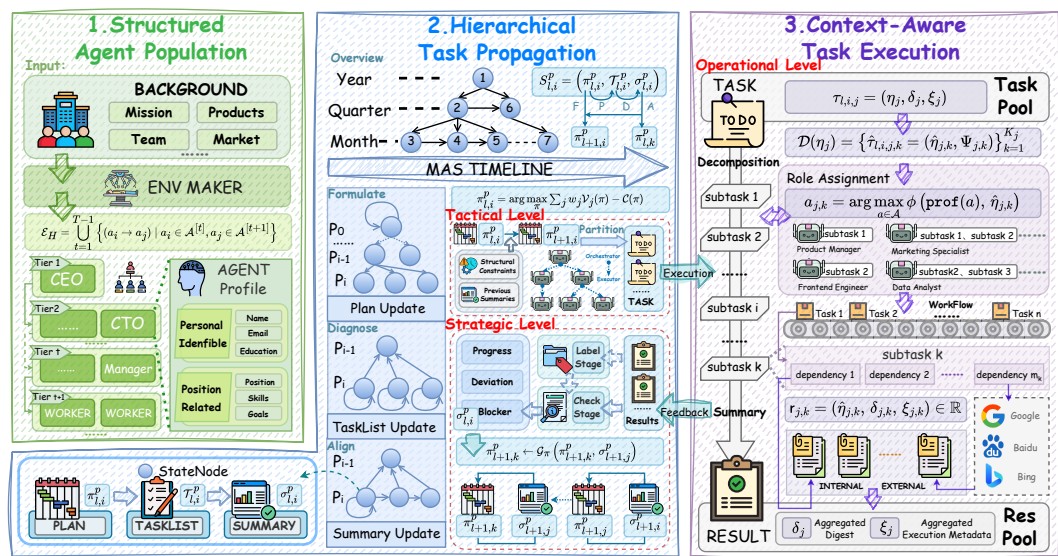

Figure 2: TaskWeave simulates enterprise dynamics with structured agents under a three-level orchestration (strategic, tactical, operational) that connects global planning with adaptive execution. The framework prepares agents from metadata, propagates intent via the FPDA cycle to generate plans and tasks, then executes them through decomposition, assignment, and external interaction.

structural constraints, We define a static delegation graph $\mathbb{H} = (\mathcal{A}, \mathcal{E}_H)$:

$$\mathcal{E}_H = \bigcup_{t=1}^{T-1} \left\{ (a_i \rightarrow a_j) \mid a_i \in \mathcal{A}^{[t]}, a_j \in \mathcal{A}^{[t+1]} \right\} \tag{3}$$

Each edge $(a_i \rightarrow a_j)$ encodes a valid delegation path from a higher-tier agent to a subordinate.

## 3.3 HIERARCHICAL TASK PROPAGATION

At the strategic level, we design a recursive mechanism that perceives internal and external information to produce high-quality enterprise plans. Inspired by control theory (Deming, 1986) and strategic management (Anand & Ward, 2002; Mintzberg, 1979), this mechanism models organizations as temporally evolving, feedback-regulated systems. We formalize the enterprise state space as $\mathbb{S}$:

$$S_{l,i}^p = \left( \pi_{l,i}^p, \mathcal{T}_{l,i}^p, \sigma_{l,i}^p \right), \quad \pi_{l,i}^p = \arg\max_{\pi} \sum_j w_j \mathcal{V}_j(\pi) - \mathcal{C}(\pi) \tag{4}$$

where each state node $S_{l,i}^p$ encapsulates localized planning and execution status, with $\pi$ denoting strategic intent, $\mathcal{T}$ a structured task bundle, and $\sigma$ a performance summary derived from execution outcomes. Nodes are arranged across temporal levels $l$ and linked to parent nodes $p$ for top-down intent inheritance and lateral adaptation to simulate long-horizon planning.

**Planning Objective.** As defined in Equation equation 4, each plan $\pi_{l,i}^p$ across the temporal hierarchy is optimized by balancing multiple utility objectives against execution cost. While we do not compute this objective function explicitly, we use it to guide prompt construction—encouraging LLM agents to implicitly weigh alignment, value, and feasibility during plan generation. Here, $\mathcal{V}_j(\pi)$ denotes the value of plan $\pi$ with respect to objective $j$, $w_j$ is the assigned weight, and $\mathcal{C}(\pi)$ denotes cost.

**Recursive Propagation.** Enlightened by foundational ideas in classical quality-control theory, we introduce a multi-agent-compatible reformulation: the **Formulate–Partition–Diagnose–Align (FPDA)** cycle. FPDA extends traditional feedback loops such as PDCA (Plan–Do–Check–Act) (Johnson, 2002) into a recursive substrate for hierarchical task propagation, lateral coordination, and adaptive refinement. It ultimately **bridging** strategic planning with fine-grained operational execution.

Each node $S_{l,i}^p$ initiates propagation as follows:

$$\begin{cases} S_{l+1,j}^p = \mathcal{F}_{\text{Align}} \circ \mathcal{F}_{\text{Diagnose}} \circ \mathcal{F}_{\text{Partition}} \circ \mathcal{F}_{\text{Formulate}} \left( S_{l,i}^p, \mathcal{A}^{[t]} \right) & \text{(Downward)} \\ \pi_{l+1,k}^p \leftarrow \mathcal{G}_\pi \left( \pi_{l+1,k}^p, \sigma_{l+1,j}^p \right), \quad \forall k \in \mathcal{N}_{\text{sibling}}^+(j) & \text{(Lateral)} \end{cases} \quad (6)$$

The proposed FPDA operator encapsulates the following control phases:

- **Formulate:** Derives a localized subgoal $\pi_{l+1,j}^p$ from the parent plan $\pi_{l,i}^p$, conditioned on current enterprise context, structural constraints, and historical performance summaries $\sigma_{l,i}^p$.

- **Partition:** Translates the subgoal into an executable task list $\mathcal{T}_{l+1,j}^p$, and injects them into the global task pool $\mathbb{T}$ for role-aware assignment and distributed processing (see Section 3.4).

- **Diagnose:** Aggregates execution signals and results $\mathbb{R}$ from downstream agents to produce a performance summary $\sigma_{l+1,j}^p$ that reflects task progress, deviations from intent, and latent blockers.

- **Align:** Applies the adaptive function $\mathcal{G}_{\text{Align}}$ to synchronize sibling nodes, leveraging diagnostic signals for lateral coordination and fine-tuned plan correction.

At the tactical level, agents dynamically alternate between orchestrator and executor based on tiers. Given a propagated subgoal $\pi_{l,i}^p$ and diagnostic summary $\sigma_{l,i}^p$, the tactical tier refines them into actionable subgoals and partitions them into task bundles $\mathcal{T}_{l+1}^p$ for downstream execution. This enables top-down delegation and peer coordination, with the tactical layer refining and integrating.

## 3.4 CONTEXT-AWARE TASK EXECUTION

At the operational level, drawing on division-of-labor theory (Becker & Murphy, 1992), we design a pipeline that turns high-level plans into agent actions. Rather than open-ended negotiation (Cemri et al., 2025), TaskWeave decomposes tasks into role-aligned subtasks, delegating them along organizational chains, and augmenting execution with historical task retrieval and external tool interactions.

**Decomposition.** For a composite task $\tau_{l,i,j} = (\eta_j, \delta_j, \xi_j)$ in state node $S_{l,i}$, where $\eta_j$ includes `background`, `description` and `constraints`. A deterministic operator $\mathcal{D}$ decomposes it into a set of atomic subtasks:

$$\mathcal{D}(\eta_j) = \{\hat{\tau}_{l,i,j,k} = (\hat{\eta}_{j,k}, \Psi_{j,k})\}_{k=1}^{K_j} \quad (5)$$

where $\hat{\eta}_{j,k}$ denotes the $k$-th atomic goal, and $\Psi_{j,k}$ encodes dependency queries.

**Role-Based Dispatch.** Each atomic subtask $\hat{\tau}_{j,k}$ is assigned to an agent $a_{j,k} \in \mathcal{A}^{[t]}$ within the target execution tier $t$, selected based on profile compatibility and organizational delegation structure:

$$a_{j,k} = \arg\max_{a \in \mathcal{A}^{[t]}} \phi\left(\texttt{prof}(a), \hat{\eta}_{j,k}, \rho_a\right) \quad (6)$$

where $\phi$ scores the alignment between the agent's role and profile (e.g., expertise, prior outputs, functional scope) and the semantic intent of $\hat{\eta}_{j,k}$. The selection is constrained to tier $t$ as determined by the delegation origin (typically from agents in $\mathcal{A}^{[t-1]}$) and structural routing rules in $\mathbb{H}$.

**Tool-Augmented Contextual Execution.** Before execution, each agent $a_{j,k}$ interprets its assigned subtask $\hat{\tau}_{j,k}$ and resolves $\Psi_{j,k}$ by retrieving a contextual bundle $\Gamma_{j,k}$ from the result pool $\mathbb{R}$, querying external tools. These tool-mediated actions enable agents to access external environments (e.g., databases, search engines, or cloud services) while grounding outputs in prior results. The agent then performs a context-grounded reasoning step on $(\hat{\eta}_{j,k}, \delta_{j,k}, \xi_{j,k})$, yielding an output that integrates historical knowledge, task semantics, and situational dependencies. Formally, the execution yields:

$$\mathrm{r}_{j,k} = (\hat{\eta}_{j,k}, \delta_{j,k}, \xi_{j,k}) \in \mathbb{R} \quad (7)$$

where $\delta_{j,k}$ is the model-generated output, and $\xi_{j,k}$ encodes provenance metadata which is stored in $\mathbb{R}$.

**Result Integration.** Once all atomic results $\{\mathrm{r}_{j,k}\}_{k=1}^{K_j}$ are completed, they are merged to update the composite task $\tau_{l,i,j}^p$, producing an aggregated digest $\delta_j$ and aggregated execution metadata $\xi_j$, both of which are stored in the result pool $\mathbb{R}$ for future retrieval and propagation.

## 4 EXPERIMENTS

### 4.1 SIMULATION SETUP

From Sec.4.2 to Sec.4.7, we evaluate *TaskWeave* through a year-long simulation of an example SaaS company, **CompanyA**. The organization is instantiated with **14 role-specialized agents** distributed across **3 hierarchical tiers** (boss, managers, workers) and spanning three functional domains—**Technology**, **Marketing**, and **Strategy**. This setup mirrors real-world enterprise structure, enabling strategic, managerial, and operational delegation. The simulation unfolds along a four-level temporal hierarchy (year → quarter → month → week), supporting long-horizon planning and short-horizon execution. Under this CompanyA background, we analyze all key modules of TaskWeave, including task generation, role assignment, plan propagation, execution dynamics, downstream utility, and interaction with external environments. To assess backbone influence, we instantiate the framework with six LLMs—`GPT-4o-mini`, `Gemini-2.0-Flash`, `Deepseek-v3`, `Moonshot-v1-8K`, `LLaMA3.1-70B`, and `GLM-4-Flash`, while keeping parameters fixed for comparability. Beyond CompanyA, we further test *generalizability* across three distinct settings in Sec. 4.8. Furthermore, we conduct an ablation analysis on each level in Appendix E, with details on human and LLM-as-a-judge evaluations, both reliably validated, in Appendix F.

### 4.2 TASK GENERATION DYNAMICS

At the strategic level, we evaluate whether TaskWeave improves task generation quality compared with single-agent baselines. The focus is on producing balanced, diverse, and high-quality.

**Settings.** We assess generation along three axes: (i) *functional balance*, measured by KL divergence against an expert-defined distribution (Tech 40%, Mkt 40%, Strat 20%), which serves only as a balanced standard rather than the unique ratio;; (ii) *semantic diversity*, computed via pairwise embedding similarity with `all-mpnet-base-v2` (H. Face, 2024); and (iii) *task criticality*, judged against a human-curated list of essential tasks. In addition, we rate realism, logical soundness, and actionability on a 1–10 scale using both domain experts and LLMs.

Table 1: Task Evaluation across 6 backbones and single-agent baselines. Total (task count), Tech/Mkt/Strat (functional allocation), KL-D (distributional balance), Div. (semantic diversity), Core-M (Core tasks by model judgment), and Core-H (by human judgment), Appendix M).

| Model | Method | Total | Tech | Mkt | Strat | KL-D | Div. | Core-M | Core-H |
|---|---|---|---|---|---|---|---|---|---|
| ChatGPT | Single | 300 | 40.33% | 27.33% | 32.34% | 0.0788 | 35.22 | 5.67% | 7.33% |
| | Ours | 228 | 16.67% | 68.42% | 14.91% | 0.2561 | 48.52 | 72.81% | 81.14% |
| Gemini | Single | 300 | 60.00% | 20.00% | 20.00% | 0.1510 | 32.59 | 3.00% | 8.67% |
| | Ours | 255 | 43.53% | 32.16% | 24.31% | 0.0203 | 55.85 | 37.25% | 78.03% |
| Deepseek | Single | 300 | 51.00% | 20.00% | 29.00% | 0.1342 | 55.52 | 13.33% | 18.67% |
| | Ours | 241 | 42.74% | 30.29% | 26.79% | 0.0323 | 57.20 | 48.55% | 92.12% |
| Moonshot | Single | 300 | 50.34% | 28.00% | 21.66% | 0.0478 | 40.03 | 8.33% | 11.00% |
| | Ours | 128 | 31.25% | 50.78% | 17.97% | 0.0358 | 42.87 | 66.41% | 93.75% |
| LLaMa3 | Single | 100 | 77.00% | 2.00% | 21.00% | 0.5423 | 32.74 | 8.00% | 10.67% |
| | Ours | 79 | 43.04% | 30.38% | 26.58% | 0.0340 | 39.35 | 37.97% | 86.08% |
| GLM | Single | 300 | 57.00% | 23.00% | 20.00% | 0.1076 | 41.72 | 9.33% | 12.33% |
| | Ours | 299 | 17.79% | 31.87% | 50.34% | 0.3579 | 54.58 | 37.46% | 76.92% |

**Results.** Across all LLM backbones, TaskWeave yields more balanced task allocations, greater semantic diversity, and higher coverage of critical managerial tasks. Quality scores remain above 7.0 for all models, indicating that the generated tasks are coherent and actionable.(detailed in Appendix H)

**Insight.**① *Coordinated MAS can generate balanced, diverse, and strategically aligned tasks with high quality, making them powerful engines for realistic enterprise simulation.*

### 4.3 ROLE ASSIGNMENT DYNAMICS

At the *tactical level*, we examine whether agents distribute work across all roles rather than concentrating execution on a few. Using the enterprise background, we curated an ideal distribution over 14 positions (Appendix G.2, a balance benchmark as above, not the only standard)), and measured divergence between observed assignments and this reference profile via KL divergence.

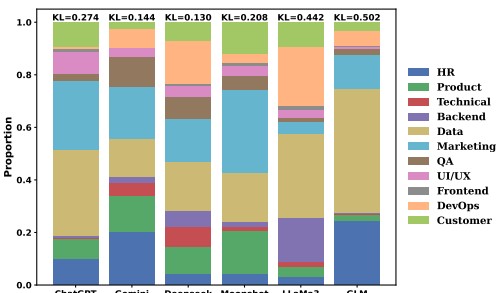

Figure 3: Role assignment distribution across different backbone models. Each bar shows the proportion of tasks executed by each role and the KL divergence above each column.

**Insight.② *Role activation in MAS reflects the backbone model's generative preferences and its capacity for balanced coordination. Weaker models(e.g., GLM) tend to overuse certain roles, while stronger ones(e.g., Gemini, Deepseek) achieve more structured coverage, showing that delegation fidelity is constrained by both orchestration design and base model expressiveness.***

### 4.4 TASK PROPAGATION DYNAMICS

Across strategic and tactical layers, we evaluate whether hierarchical plans in TaskWeave are effectively coordinated testing if high-level intents are properly decomposed and refined under the FPDA mechanism. For each plan, an external LLM (Gemini-2.5-Flash) generates a checklist of actionable checkpoints, and we measure completion rates from execution summaries, considering both current-phase progress and cumulative resolution under delayed execution (e.g., two months for monthly plans). This simulates temporal spillover in enterprise workflows, with details in Appendix **??**.

As shown in Table 2, hierarchical plans are progressively decomposed and realized across monthly and weekly layers. While some models struggle to complete monthly tasks, subgoals are often finished in later cycles, showing sustained intent and coherent progression. Weekly plans achieve higher completion due to finer granularity.

Table 2: Completion rate across different backbone under monthly and weekly evaluation. '-' means that the plan propagates poorly

| Model | Monthly | | | Weekly | | |
|---|---|---|---|---|---|---|
| | n-check | Timely | Finalized | n-check | Timely | Finalized |
| ChatGPT | 30 | 76.67% | 91.39% | 14.25 | 71.35% | 95.32% |
| Gemini | 29.58 | 81.92% | 92.66% | 14.41 | 87.28% | 98.58% |
| Deepseek | 27.8 | 75.18% | 82.01% | 13.41 | 78.88% | 92.55% |
| Moonshot | 24.3 | 71.03% | 83.79% | 13.08 | 72.61% | 91.72% |
| LLaMa3 | – | – | – | – | – | – |
| GLM | – | – | – | 11.53 | 57.80% | 68.21% |

**Insight.③ *Enterprise simulation ultimately hinges on the capabilities of the backbone LLMs. When models lack reasoning ability and generative capacity, they struggle to sustain complex workflows, leaving strategic goals unrealized. Thus, the fidelity of MAS simulation is bounded not only by orchestration design but also by the fundamental competence of the models that enact it.***

### 4.5 TASK EXECUTION DYNAMICS

At the *operational level*, we evaluate TaskWeave's ability to execute interdependent tasks under evolving plans. Agents retrieve historical context, use tools for reasoning, and produce emergent execution chains. Figure 4 illustrates a workflow generated without predefined scripts, while degree analysis of the task-dependence graph (Appendix M) highlights core initiators and coordination hubs.

**Insight.④ *Agents can simulate enterprise emergency operations by generating structured plans, coordinating realistic workflows, and producing varied outcomes that mirror the execution dynamics.***

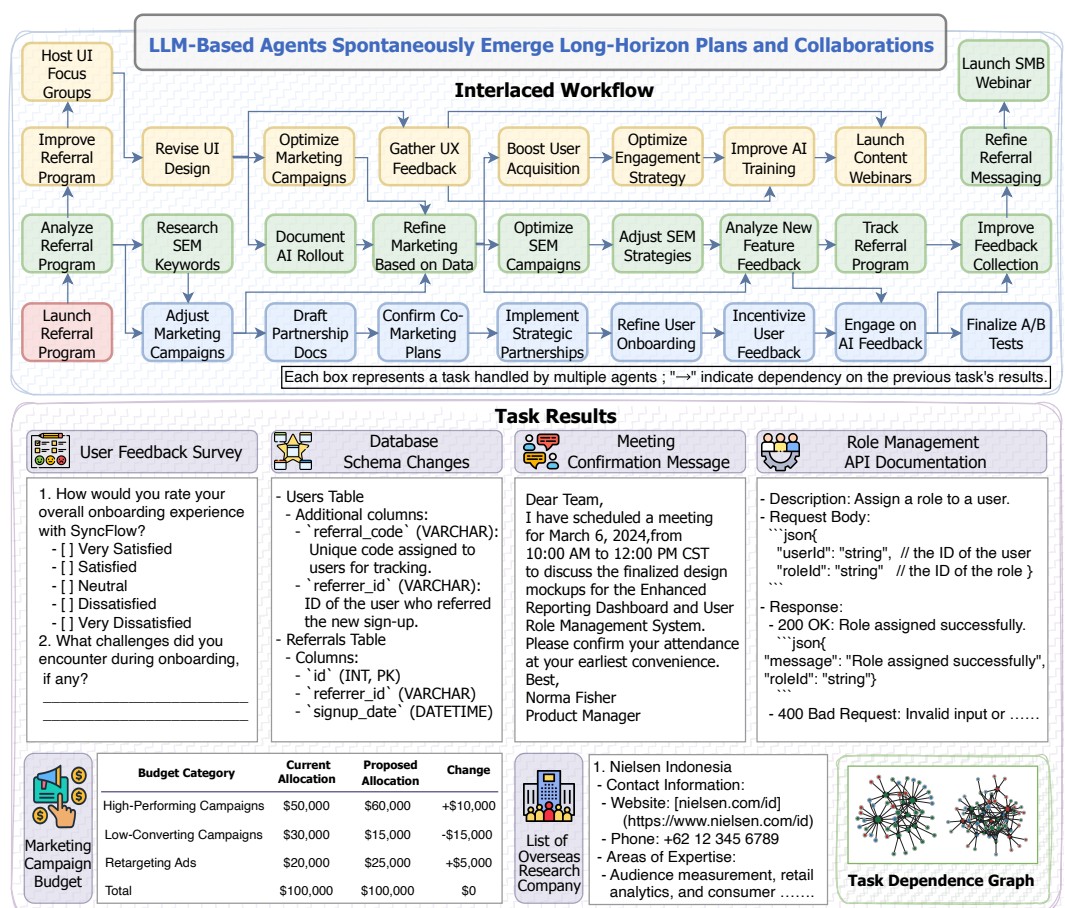

Figure 4: Case study. LLM agents generate hierarchical, temporally dependent workflows with diverse structured outputs; the dependency graph (bottom right) shows inter-task relations.

### 4.6 DOWNSTREAM UTILITY

To demonstrate the utility of generated data, we propose **Organizational Sensitive Span Detection (OSSD)**, a NLP task motivated by compliance needs (e.g., GDPR (European Union, 2016), EU Data Act (Union, 2023)). OSSD detects spans in textual data that contain sensitive information and assigns each a category and rationale. Formally, given a text $T$, the model outputs $S = \{(t_i, c_i, r_i)\}_{i=1}^{k}$, where $t_i$ is a span, $c_i$ a label (e.g., `Financial Data`), and $r_i$ a rationale. Unlike standard NER, OSSD supports enterprise-specific boundaries and categories and explicitly requires $r_i$ for compliance. **Annotation Pipeline.** We design a multi-stage *label-and-verify* process with `GPT-4o-mini`: (1) extract spans, (2) generate justifications, and (3) verify via. Human reviewers performed consistency checks on stages (2)–(3), both reliability and fairness of the annotation process (Appendix K).

**Settings.** We use 3 baselines: `MoA` (Wang et al., 2025), `Megenic-One` (Fourney et al., 2024), and `G-Designer` (Zhang et al., 2025), all deployed on the same scenario (CompanyA) with 26 tasks across 3 departments under identical agent roles. Evaluation uses API calls (API), span count (Spans), avg. length (Len), tokens (P/C), and label diversity (Div., span types ≥3).

Table 3: Internal sensitive span quantity and usage.

| Method | API | Spans | Len | Tokens (P/C) | Div. |
|---|---|---|---|---|---|
| MoA | 14.1 | 454 | 34.77 | 27k/5k | 11 |
| Megenic-One | 31.0 | 1538 | 52.04 | 465k/19k | 83 |
| G-Designer | 40.6 | 252 | 28.22 | 16k/25k | 19 |
| Ours | **5.6** | **643** | **37.78** | **30k/4k** | **31** |

**Results and Analysis.** As shown in Table 3, while all methods can generate some internal-sensitive spans, TaskWeave yields richer spans (643) with lower API cost (5.6 calls per task) and higher token efficiency. To further validate the *non-triviality* of the OSSD task, we evaluated 300 short spans

(length $\leq 10$) using two advanced NER models (Zhou et al., 2024; Ding et al., 2024), and observed F1-score drops of over 40%, highlighting the intrinsic distinction between our generated textual spans and publicly available entity types. Furthermore, a file-level sensitive data analysis in Appendix I.

**Insight.⑤** *By simulating the realistic dynamics, MAS frameworks can enrich enterprise simulations with semantically grounded content at lower cost, while also supporting sensitive synthetic data generation by emulating the processes behind sensitive information rather than accessing it directly.*

### 4.7 Interacting with External Environments

Beyond internal various dynamics, we evaluate TaskWeave under two complementary settings: (i) injecting external incidents into the strategical level (ENV → MAS), and (ii) agents interact with environment via tools (MAS → ENV).

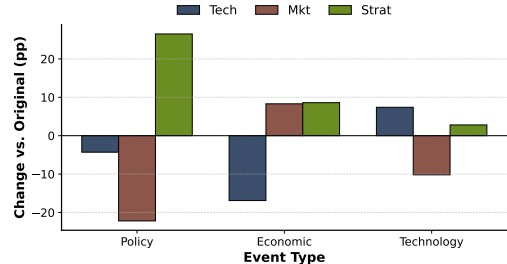

**External Incidents.** We injected 15 realistic events for 3 types into quarterly planning of CompanyA. The system adaptively shifted workloads—policy shocks emphasized strategic planning, while economic shocks increased market tasks (Fig. 5).(detailed in Appendix N.1).

Figure 5: Impact of external incidents.

**Tool Interaction.** Equipped with 63 tools spanning SQL, social-media, and office, agents completed a one-month simulation with 534 tool calls (41.3% of steps). Office tools dominated usage, SQL enabled centralized queries, and media tools supported communication.(detailed in Appendix N.2). **Insight.⑥** *LLM agents in MAS can imitate enterprise dynamics by adapting to external perturbations and leveraging tools, thereby realistically bridging organizational behavior with its environment.*

### 4.8 Generalizability Across Domains

We evaluate TaskWeave across three other organizational settings: a financial company (**Fin**), a manufacturing plant (**Manu**), and a government agency (**Gov**) for one month(detailed settings in Appendix.L. As shown in Table 4, the framework generalize across different backgrounds, larger sizes, deeper hierarchies, and more functional departments while sustaining stable task completion. With increasing organizational complexity, task diversity and agent activations also rise, indicating stronger adaptability.

Table 4: Generalizability experiments. **Size** (number of agents), **Tiers** (hierarchical depth), **Dep.** (functional departments), **Task** (executed tasks), **Div.** (task diversity index), **Role** (agent activations), **Com.** (task completion rate), **Spans** (OSSD spans labeled in 50 tasks).

| Scene | Size | Tiers | Dep. | Task | Div. | Role | Com. | Spans |
|-------|------|-------|------|------|------|------|------|-------|
| Fin | 15 | 3 | 3 | 20 | 45.05 | 119 | 81.6% | – |
| Manu | ~50 | 4 | 4 | 129 | 54.36 | 655 | 88.0% | – |
| Gov | ~100 | 5 | 5 | 290 | 58.55 | 1782 | 86.0% | 1651 |

adaptability. In government case, 1,651 sensitive spans further highlight the value of TaskWeave for downstream data synthesis, demonstrating both robustness in execution and utility in data synthesis. **Insight.⑦** *By capturing the core features of enterprise dynamics, MAS frameworks can generalize across domains and remain robust under growing complexity, while higher agent activations reveal the growing cost and complexity of coordination.(as reflected by **Com.** and **Role**)*

## 5 Conclusion

This paper presents *TaskWeave*, a multi-agent framework that simulates enterprise dynamics by orchestrating LLM-based agents across three interlocking layers: strategic planning, tactical delegation, and operational execution. By aligning role specialization with temporal abstraction, TaskWeave addresses key challenges in modeling structural complexity, long-horizon planning, and context-grounded execution. Experiment results show its ability to support faithful, scalable simulations that reduce the overhead of studying organizational behavior. Our insights suggest that TaskWeave can serve as a valuable foundation for data synthesis, enterprise modeling, and decision support in dynamic environments. Future work includes the deployment of our TaskWeave in more ecosystems.

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

## A   LLM USAGE

In preparing this work, we made use of large language models (LLMs) to assist with specifictasks. LLMs were used for linguistic refinement (grammar and style polishing) and related research exploration. All model outputs were carefully reviewed by the authors to ensure factual accuracy and faithfulness to the intended meaning of our work. We explicitly verified that no fabricated references, hallucinated claims, or misleading statements were introduced. The scientific content, analysis, and contributions remain solely the responsibility of the authors.

## B   ETHICS STATEMENT

Our experiments involve the generation of synthetic enterprise data, which may contain elements resembling company backgrounds or personal information. We emphasize that all such data are entirely artificial and constructed for simulation purposes only. No real organizations, employees, or proprietary records are included, and the synthetic nature of the data ensures that it does not impact or compromise any real individuals or enterprises. We believe that this work poses no risk of privacy infringement or misuse of sensitive information.

## C   OPEN RESOURCE

Our code and generated data can get from: `https://anonymous.4open.science/r/ TaskWeave-F0F6`.

## D   COMPARISON

In this section, we provide additional comparisons of *TaskWeave* against (i) traditional enterprise modeling tools such as BPMN and process mining systems, and (ii) existing LLM-based MAS frameworks. These materials complement our discussion in the main text and highlight the unique positioning of TaskWeave.

### D.1   COMPARISON WITH BPM METHODS

Traditional enterprise modeling tools, such as *Business Process Management Notation* (BPMN) simulators and process mining engines, have been widely adopted to model organizational workflows. However, these tools typically rely on static process templates and pre-defined control logic, making them less effective in environments that require continuous adaptation to shifting goals, unexpected events, or external shocks.

By contrast, TaskWeave complements such tools by enabling **open-ended, hierarchical simulation** through LLM-based agents. In particular, compared to template- or log-driven approaches, TaskWeave provides:

- **Higher fidelity:** It captures both formal and informal coordination via dynamic intent propagation, rather than relying solely on fixed schemas or historical logs.
- **Greater flexibility:** It supports adaptive task generation and multi-agent workflows without pre-defined process templates.
- **Practical efficiency:** Despite higher token usage, TaskWeave reduces manual modeling effort and runs effectively on low-cost LLM backbones.

We further conducted qualitative case analyses, summarized below, with detailed discussions deferred to the Appendix.

### D.2   COMPARISON WITH EXISTING MAS

Prior LLM-based MAS frameworks (Hong et al., 2024; Park et al., 2023; Chen et al., 2025) have demonstrated strong capabilities in collaborative task-solving, sandboxed simulations, and meta-evaluation of generative agents. However, most emphasize local planning or isolated task execution,

Table 5: Qualitative comparison of BPMN and TaskWeave

| Aspect | BPMN | TaskWeave |
|---|---|---|
| Process Structure | Static | Dynamic |
| Adaptability | Low | High |
| Error Handling | Predefined | Dynamic |
| Context Integration | Limited | Supported |
| Domain Portability | Limited | Supported |
| Simulation Capability | Descriptive | Executable |

with limited centralized coordination or temporal hierarchy. As a result, they often fall short in modeling long-horizon enterprise workflows with continuity, dependencies, and adaptive feedback.

TaskWeave is not a generic MAS or dialog agent framework, but a structured system tailored for enterprise modeling and data synthesis. It uniquely integrates hierarchical planning, adaptive control loops, and traceable execution, enabling end-to-end simulation of organizational processes at scale. Table 6 presents a conceptual comparison.

Table 6: Conceptual comparison of TaskWeave and prior LLM-based MAS frameworks across eight key dimensions. Symbols: ✓ supported, ✗ not supported, ▲ partially supported.

| Framework | Sim | Hier | Goal | Long | Loop | Trace | Auto | Vers |
|---|---|---|---|---|---|---|---|---|
| **TaskWeave** | ✓ | ✓ | ✓ | ✓ | ✓ | ✓ | ✓ | ✓ |
| CAMEL | ✗ | ✗ | ▲ | ✗ | ✗ | ✗ | ▲ | ✓ |
| Autogen | ✗ | ▲ | ✓ | ✗ | ✗ | ▲ | ✓ | ✓ |
| GPTSwarm | ✗ | ▲ | ✓ | ✗ | ✓ | ▲ | ✓ | ✓ |
| MacNet | ✗ | ▲ | ✓ | ✗ | ✗ | ▲ | ✓ | ✓ |
| GenAgents | ▲ | ✗ | ✗ | ✓ | ✗ | ✓ | ✓ | ✓ |
| MetaGPT | ▲ | ✓ | ✓ | ✗ | ▲ | ✓ | ✓ | ✗ |
| TwinMarket | ✓ | ✗ | ✗ | ✓ | ✗ | ✓ | ✓ | ▲ |
| SOCIODOJO | ✓ | ✗ | ▲ | ✓ | ▲ | ✓ | ✓ | ▲ |
| VIRSCI | ▲ | ✗ | ▲ | ✗ | ✗ | ✓ | ✓ | ✗ |

## E  ABLATION ANALYSIS

We verify the contribution of each level through targeted empirical analyses. The *strategic level* (Section 4.2, 4.4) ensures long-horizon coherence planning and adaptive refinement. The *tactical level* (Section 4.3, 4.4) promotes inclusive collaboration and balanced role engagement. The *operational level* (Section 4.5, 4.6) enables context-aware execution with traceable, high-utility outputs. Together, these results validate that each level enhances TaskWeave's fidelity, adaptability, and coherence.

## F  RELIABILITY OF EVALUATION METHODS

Given the novelty of our LLM-based MAS simulation setting, some open-ended evaluations necessarily involve human and LLM-based judgments. We followed best practices from prior MAS simulation worksPark et al. (2023) and designed structured evaluation method to ensure rigor and transparency. The key evaluation dimensions and protocols are summarized as follows.

**LLM-as-a-judge.** As described in Section 4, we carefully designed prompts with three categories, nine criteria, and dual-descriptor definitions (see Appendix M). We further validated reliability via Cohen's Kappa over 100 repeated samples, obtaining a high agreement score of 0.88. In Section 4.4, we implemented a two-stage pipeline consisting of (a) generation and (b) reasoning and tracing, and verified consistency between two independent LLMs. An ablation study was also conducted (Appendix F). For Section 4.6, we applied a three-stage labeling pipeline ( K).

**Human Expert.** We involved four experts (A–D): Expert A has over 8 years of management experience in IT companies; B–D have 1–2 years of experience in development/operations, all

holding a master's degree or above. All experts underwent standardized training before conducting evaluations.

Table 7: Evaluation Methods and Reliability Measures

| Method | Location in Paper | Quality Assurance Strategy |
|---|---|---|
| LLM-as-a-judge | Section 4.2 | Crafted prompts with 3 categories, 9 criteria, and dual-descriptor definitions (Appendix D). Cohen's Kappa over 100 repeated samples (score = 0.88). |
| | Section 4.4 | Two-stage pipeline: (a) generation; (b) reasoning and tracing. Consistency between two LLMs. Ablation study (Appendix J). |
| | Section 4.6 | Three-stage labeling pipeline (Appendix K ). |
| Human Expert | Section 4.1 / 4.3 | Expert A designed; distributions represent one realistic configuration (Appendix M and Appendix G). |
| | Section 4.2 | Based on semantic clustering; Expert A refined the final list and Experts B, C, D classified (Appendix M). |
| | Section 4.6 | 500 samples manually checked by Experts B, C, D; $\sim$5–10% noise retained to reflect LLM imperfection (Appendix K). |

## F.1

# G  AGENT POPULATION

## G.1  ROLE-EXAMPLE

To achieve a more realistic and credible simulation effect, we utilize customized prompts to comprehensively define each character by incorporating both personal identifiers (e.g., name, email, education, etc.) and role-specific attributes (e.g., goals, constraints, etc.) during the construction of various agents. To enhance comprehension, the following two representative prompt examples are presented. The first example is from the managerial strata, while the other is from the operational strata.

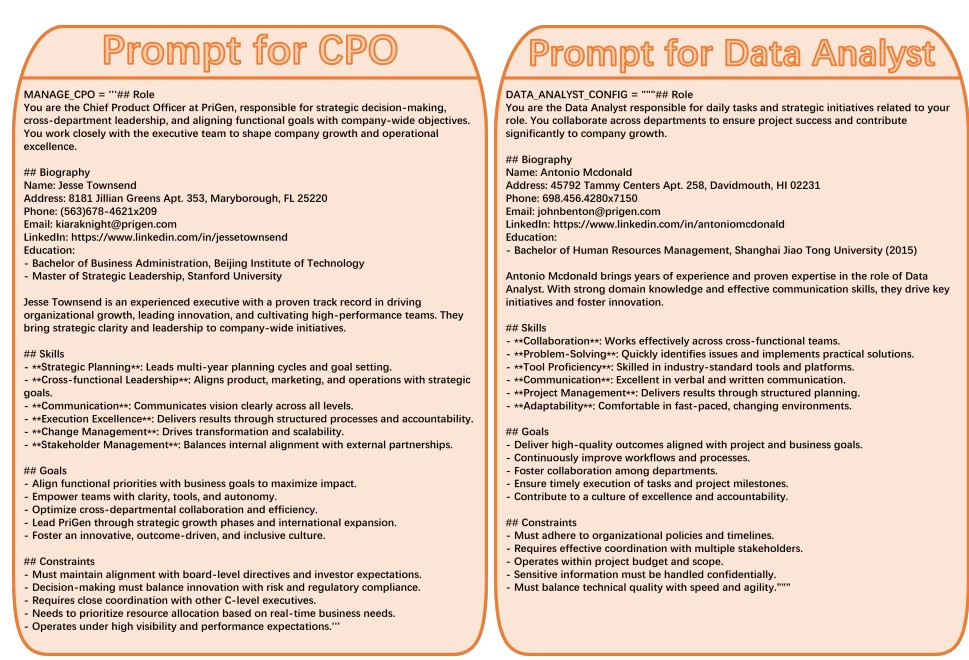

Figure 6: Prompt Examples.

## G.2 IDEAL ROLE DISTRIBUTION

We invite experts to design roles according to their company background. The job distribution is as follows.

| Role | Proportion |
|------|-----------|
| HR | 0.12 |
| Product Manager | 0.15 |
| Technical Support Engineer | 0.05 |
| Backend Engineer | 0.05 |
| Data Analyst | 0.15 |
| Marketing Specialist | 0.15 |
| QA Engineer | 0.05 |
| UI/UX Designer | 0.05 |
| Frontend Engineer | 0.05 |
| DevOps Engineer | 0.05 |
| Customer Success Manager | 0.10 |

Table 8: Expert-defined ideal role distribution for the simulated SaaS enterprise.

## H TASK QUALITY EVALUATION

### H.1 TASK DESCRIPTION QUALITY

We used a comprehensive evaluation of task quality to ensure the reliability and validity of the results generated. The evaluation process integrated both automated and human evaluations to obtain a multifaceted and objective quality analysis.

The task quality evaluation was primarily focused on the core criteria in Table 9.

During the process of the model evaluation, we used two models —— ChatGPT-4o-mini and Gemini-2.0-flash, to evaluate task quality according to the descriptions of different metrics in the *Task Generation Evaluation Metric System*. The result of model evaluation shows that the final tasks generated using different models in our framework have similar quality and all obtain a high score of more than 7 out of 10. Meanwhile, these scores from both models have a consistent gradient of change and the phenomenon is shown in Figure 7.

This result is basically the same as the result of human evaluation, that is, the quality of different tasks generated is similar and relatively high. This demonstrates that PriGen, the company employing framework simulation, possesses strong operational capabilities, high usability, and proven authenticity.

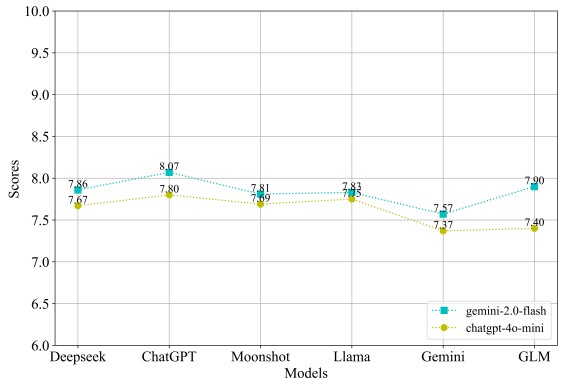

Figure 7: Task Quality Evaluated by Model. The score range is set [0, 10]. Obviously, all generated tasks have scores above 7, which means that all generated tasks are of high quality.

| Dimension | Metric Name | Concise Meaning |
|---|---|---|
| Realism | Factuality | 1. Does the generated task content accurately reflect the real company background and business situation?
2. Is the key information in the task description consistent with known facts? |
| | Industry Relevance | 1. Does the generated task align with the standard operations and norms of the target industry or domain?
2. Does the task content match the business processes of the relevant industry? |
| | Detail Realism | 1. Are the details in the task description specific and reasonable?
2. Does it include practical timelines, budget estimates, and assigned roles? |
| Logicality | Completeness | 1. Does the task decomposition cover all necessary aspects to achieve the original goal?
2. Does the sub-task list include all crucial information and steps required for the overall objective, without omissions? |
| | Coherence | 1. Is the logical order and structure between tasks and sub-tasks reasonable and smooth?
2. Does the task list have clear relationships of sequence and causality, avoiding abrupt shifts or irrelevant content? |
| | Non-Redundancy | 1. Are there any duplicate, conflicting, or irrelevant sub-tasks in the task list?
2. Does the task avoid redundant steps or contradictory requirements? |
| Actionability | Feasibility | 1. Is the generated task plan practically implementable in reality?
2. Can each sub-task be completed by a specific role or team within the given resource and time constraints? |
| | Actionability | 1. Is the task presented in a clear, actionable format? Does it include specific action verbs, subjects, and steps?
2. Does the task clearly state "who does what"? |
| | Resource Clarity | 1. Is the information regarding resource needs, timelines, and deliverables sufficiently clear in the task plan?
2. Does it explicitly mention required roles, budget estimates, tools, or processes? |

Table 9: Task Generation Evaluation Metric System

## H.2 GENERATE TASKS WITH SINGLE AGENT

When modeling the company, we carefully construct the company background, which is shown in Figure 8, to simulate the company situation in the real environment. In the background, we define seven essensial parts: Company Overview, Company Mission & Values, Products & Services, Market & Target Audience, Technology & Data Management, Operational Data & Internal Processes and Team Culture & Growth Plans. These settings cover all the key information that companies care about in the real world.

**Company Overview**
PriGen is an innovative SaaS startup founded in 2019, based in Shanghai's Zhangjiang Hi-Tech Park. The company was established by former engineers from leading tech companies like Tencent and Google, along with a strong product and marketing team. PriGen's primary offering, SyncFlow, integrates all essential tools for remote teams into one unified platform. From project management, team communication, document collaboration, and analytics, SyncFlow simplifies operations and fosters teamwork, particularly for small to medium-sized businesses (SMBs) that need an integrated suite of tools.

**Company Mission & Values**
PriGen's mission is to enable seamless collaboration for teams by providing an all-in-one platform that integrates real-time document collaboration, task management, communication, and analytics. The company values customer-centricity, continuous innovation, and operational efficiency. PriGen's core values emphasize simplicity, scalability, and adaptability. By focusing on delivering a flexible and easy-to-use product, PriGen aims to simplify the complexities of managing remote teams.

**Products & Services**
SyncFlow is the company's flagship product, designed to support businesses by simplifying team collaboration and project management. It combines task management, document collaboration, communication tools, project tracking, and real-time analytics into a single platform. The system is built using modern technologies like React.js and TypeScript for the frontend, FastAPI (Python) for the backend, and PostgreSQL for data storage. SyncFlow integrates with external tools like Google Drive, Slack, and Microsoft Teams to enhance the productivity of its users.

**Market & Target Audience**
SyncFlow primarily targets SMBs, especially those in the tech, marketing, and consulting industries. The platform aims to replace the inefficient and fragmented tool ecosystem many small businesses rely on. SyncFlow helps these businesses streamline their workflows, manage tasks more effectively, and ensure smoother team communication. The primary users of SyncFlow are project managers, team leads, and product designers. The company's marketing strategy includes direct sales, content marketing, social media campaigns, and partnerships with other SaaS platforms.

**Technology & Data Management**
SyncFlow is hosted on AWS, providing auto-scaling capabilities and ensuring high availability. The platform uses **AES-256** encryption for data storage and **TLS 1.3** for data in transit, guaranteeing the security of sensitive information. PriGen employs role-based access control (RBAC) to restrict access to data and resources. The company also uses automated data classification to ensure sensitive data, including customer details, transaction logs, and internal technical documentation, are securely stored in accordance with data protection regulations like GDPR.

Operational Data & Internal Processes
On average, SyncFlow handles over 2,000 documents, 8,000 tasks, and 15,000 internal messages per month, generated by users across globally. System logs retain document edit trails, task update timestamps, and communication transcripts for up to 180 days, unless manually purged. PriGen has also implemented a quarterly internal data audit protocol led by its Privacy and Compliance Officer, with participation from Legal, IT, and Product teams.
Additionally, all employees undergo semi-annual performance reviews, combining peer feedback, self-assessment, and manager evaluations stored in the HR system. These records may contain sensitive content, including personal indentifible information, employee personal goals, project-specific contributions, or anonymized 360 feedback. Customer support communications and incident response logs are archived via the Zendesk API for SLA validation and trend analysis.

**Team Culture & Growth Plans**
PriGen promotes a dynamic, flat organizational structure, encouraging innovation, creativity, and collaboration. All employees are empowered to take ownership of their work and contribute ideas. The company started with a small team but plans to expand to 100 employees by the end of 2025. Key areas for expansion include customer support, sales, and marketing. PriGen also aims to scale internationally by establishing partnerships with top consulting firms in the APAC region and securing Series C funding for future growth.

Figure 8: The background of the simulated company–PriGen.

In order to prove the strong ability of our framework to generate high-quality tasks, we carefully construct prompt (Figure 9), which is used to guide different single agents to generate tasks as a control. In the prompt, we provide exactly the same company background information and the fixed format of tasks to help the agent generate the tasks.

## H.3 KL-D

The Kullback-Leibler (KL) divergence, also known as relative entropy, is an asymmetric measure of the difference between two discrete probability distributions, $P$ and $Q$. It quantifies the information lost when $Q$ is used to approximate $P$. Mathematically, for discrete distributions, the KL divergence is defined as:

$$D_{KL}(P||Q) = \sum_i P(i) \log \left( \frac{P(i)}{Q(i)} \right)$$

It's important to note that the KL divergence is generally not symmetric, meaning $D_{KL}(P||Q) \neq D_{KL}(Q||P)$. Furthermore, if there exists an event $i$ such that $P(i) > 0$ but $Q(i) = 0$, the KL divergence becomes infinitely large. Because of these properties, the KL divergence is a crucial concept in fields like machine learning and information theory for measuring the dissimilarity between discrete probability distributions.

We employ the Kullback-Leibler (KL) divergence to quantitatively evaluate the discrepancy between the distribution of generated task types and the ideal task type distribution. $P$ presents the distribution of generated task types by models while $Q$ presents the ideal task type distribution. The tasks are

Figure 9: The prompt that we provided for single agents.

categorized into three distinct types: *Technology & Product*, *Marketing & Customer* and *Organization & Strategy*. Based on empirical findings, the ideal task proportion vector $Q$ is determined to be **4:4:2**.

### H.4 TASK DIVERSITY

Incorporating task diversity indicator during task quality evaluation can significantly improve the comprehensiveness and reliability of the assessment outcomes. Evaluating the diversity of generated tasks helps move beyond a one-dimensional perspective, enabling a more holistic assessment of the task set's applicability and completeness, and offering a more accurate reflection of its overall quality.

We quantify task diversity by measuring the semantic differences between task descriptions. Specifically, we use the "all-mpnet-base-v2" model to encode each task in one task set $\mathcal{T}$ into a semantic vector $\vec{t}_i(0 < i \leq |\mathcal{T}|)$ , and calculate $DIS$:

$$DIS = 100 \times \frac{2}{|\mathcal{T}|(|\mathcal{T}| - 1)} \sum_{i<j}(1 - \cos(\vec{t}_i, \vec{t}_j))$$

as the indicator, where larger values indicate greater semantic dispersion and thus higher task diversity.

We observed that Single Model often generates repetitive tasks. To address the impact of this phenomenon on diversity evaluation, we apply Algorithm 1 to aggregate overlapping tasks and adjust the DIS based on the task group characteristics, such as the number of tasks in each group and the total number of overlapping tasks.

### H.5 CORE TASK EVALUATION

According to the company's background, we set up a core-task list (Table 10) for priGen after a lot of research and thinking. It consists of 7 tasks, 3 of which belong to the "Technology & Product" category, 3 belong to the "Marketing & Customer" category, and 1 belongs to the "Organization & Strategy" category. The ratio of task categories in the list is set to ensure that it is consistent with the ideal distribution of task category distribution–4:4:2 (metioned in H.3).

We first filter non-repetitive tasks using a method similar to part of Algorithm 1. Notably, we observe that the number of remaining tasks in the task set generated by the Single Agent is significantly reduced after filtering. It indicates that there are a large number of duplicate tasks in the task set

---

**Algorithm 1** Calculate Task Diversity

---

**Require:** $logs$: List of log file paths
 1: $model \leftarrow$ InitializeSentenceTransformer(`'all-mpnet-base-v2'`)
 2: **for all** $log \in logs$ **do**
 3:      $tasks \leftarrow$ get_tasks($log$)
 4:      $embeddings \leftarrow model$.encode($tasks$)
 5:      $count \leftarrow$ length($tasks$)
 6:      $flags, losses \leftarrow [0] \times count, [\,]$
 7:      $diversity\_sum \leftarrow 0$
 8:      **for** $i = 0$ to $count - 2$ **do**
 9:          **for** $j = i + 1$ to $count - 1$ **do**
10:              $sim \leftarrow$ cosine_similarity($embeddings[i], embeddings[j]$)
11:              $diversity\_sum \leftarrow diversity\_sum + (1 - sim)$
12:              **if** $(flags[i] = 1 \wedge flags[j] = 1) \vee sim > 0.98$ **then continue**
13:              $group\_found \leftarrow$ **False**
14:              **for all** $group \in losses$ **do**
15:                  **if** $tasks[i] \in group \vee tasks[j] \in group$ **then**
16:                      $group$.add($tasks[i]$)
17:                      $group$.add($tasks[j]$)
18:                      $flags[i], flags[j] \leftarrow 1, 1$
19:                      $group\_found \leftarrow$ **True**
20:                      **break**
21:              **if not** $group\_found$ **then**
22:                  $losses$.append($\{tasks[i], tasks[j]\}$)
23:                  $flags[i], flags[j] \leftarrow 1, 1$
24:      $sum\_len, loss\_weight \leftarrow 0, 0$
25:      **for all** $group \in losses$ **do**
26:          $loss\_weight \leftarrow loss\_weight + \frac{\text{len}(group)}{10}$
27:          **if** len($group$) $\geq 4$ **then**
28:              $loss\_weight \leftarrow loss\_weight + 1$
29:          $sum\_len \leftarrow sum\_len + $ len($group$)
30:      **if** $\frac{sum\_len}{count} > 0.1$ **then**
31:          $loss\_weight \leftarrow loss\_weight + \left( \frac{sum\_len}{count} \times 10 \right)$
32:      $diversity \leftarrow \frac{2 \times diversity\_sum}{count \times (count-1)}$
33:      **for** $k = 1$ to len($losses$) **do**
34:          $diversity \leftarrow diversity \times 0.99$

---

generated by the Single Agent, and the conclusion is also consistent with the determination result of diversity. Next, both model-based and manual evaluations are used to determine whether a task qualified as a core task. Finally, the number of core tasks is taken as the final output.

| Task Content | Category |
| --- | --- |
| Product Development and Launch. | Technology & Product |
| Performance Monitoring and Analysis. | Technology & Product |
| Customer Experience and Success. | Technology & Product |
| Sales Strategy and Execution. | Marketing & Customer |
| Marketing Campaign Optimization and Growth. | Marketing & Customer |
| Financial Planning and Investment. | Marketing & Customer |
| Strategic Partnership. | Organization & Strategy |

Table 10: The set of the core task list.

### H.6 GENERATED TASK EXAMPLES

In this section, we show part of generated tasks from single agents (Figure 10) and *TaskWeave* (Figure 11). It is observed that even within only the first five tasks, those generated by a single agent are already repetitive. In contrast, *TaskWeave* not only effectively reduces such duplication, but also organizes tasks in a structured and coherent manner, thereby better simulating a real-world workflow.

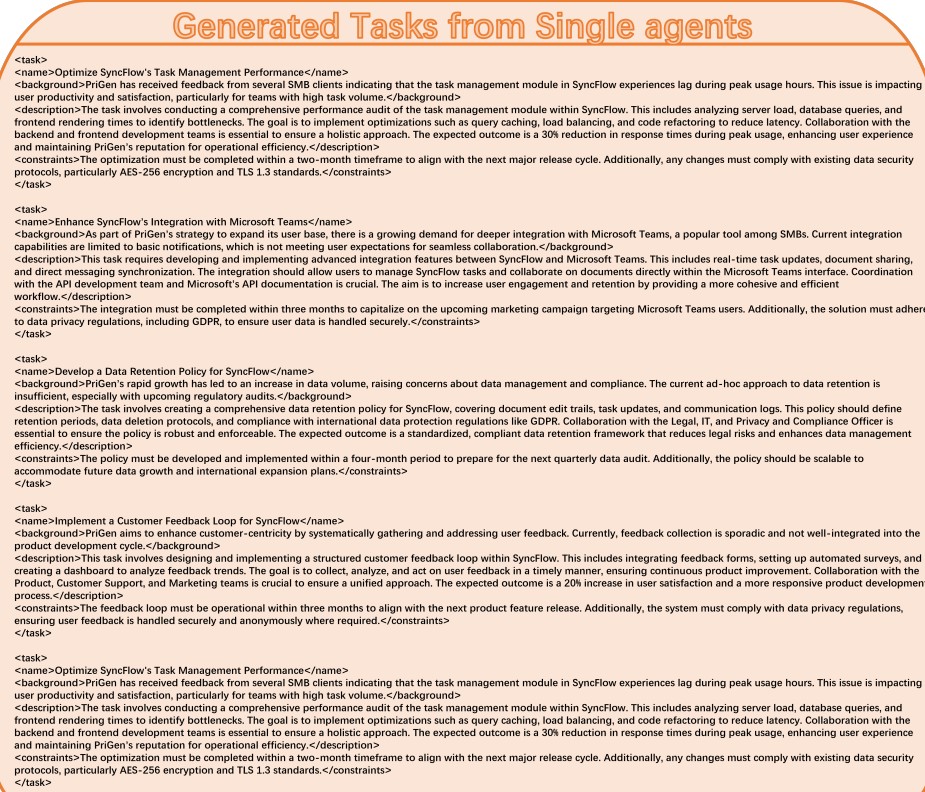

Figure 10: The generated tasks from single agents using ChatGPT-4o-mini.

## Generated Tasks from *TaskWeave*

```
<task>
<name>Refine SEM Targeting Strategy</name>
<background>PriGen's recent digital marketing campaign saw a significant increase in traffic driven by Search Engine Marketing (SEM), but conversion rates were below expectations due to broad targeting. The goal is to enhance lead quality by focusing on more relevant audiences.</background>
<description>Analyze and refine the SEM targeting strategy by exploring long-tail keywords that address specific user pain points related to PriGen's offerings. Implement geo-targeting ads in regions demonstrating strong conversion potential based on historical data. Simulate at least 100 keyword variations and their expected search volumes, along with hypothetical ad creatives designed for targeted demographics.</description>
<constraints>Ensure that the refined targeting includes a comprehensive keyword performance report that outlines the expected ROI. Generate synthetic data for keyword performance, including average CPC, search volume, and predicted conversion rate. Maintain alignment with existing campaign budgets and consider regulatory compliance concerning digital advertising standards.</constraints>
</task>

<task>
<name>Enhance Social Media Advertising Campaign</name>
<background>Social media ads contributed substantially to traffic but had low conversion rates due to ineffective targeting. Focused engagement on well-defined audience segments based on demographic and behavioral data is necessary.</background>
<description>Develop and execute a revised social media advertising plan that includes the creation of audience segments tailored to user interactions. Utilize retargeting strategies for users who previously engaged but did not convert. Design and simulate three different ad sets tailored for each segment, including creative variations and corresponding performance metrics projections.</description>
<constraints>Produce synthetic audience data representing different segments to assess potential engagement levels and conversion rates. Ensure adherence to privacy regulations regarding user data handling while testing ad performance through A/B testing protocols. Provide a summary report outlining anticipated campaign effectiveness based on simulated outcomes.</constraints>
</task>

<task>
<name>Optimize Email Marketing Campaigns</name>
<background>Email marketing has shown the lowest traffic contribution but has a better conversion rate compared to other channels. Therefore, enhancing personalization and segmentation is crucial to drive engagement and conversions.</background>
<description>Create a segmented email marketing strategy that includes A/B testing for subject lines and content tailored to different user interests. Develop a series of three email templates that reflect user behavior insights and personalize content. Simulate sending to 5,000 users across various segments to predict open rates and click-through rates based on established benchmarks.</description>
<constraints>Generate synthetic email interaction data reflecting metrics like open rates, click rates, and conversion outcomes. Ensure compliance with anti-spam laws by including appropriate unsubscribe options in all simulated email campaigns. Document and analyze A/B test results to measure effectiveness against previous campaigns.</constraints>
</task>

<task>
<name>Create Targeted Landing Pages for Content Marketing</name>
<background>The content marketing efforts have underperformed in converting traffic due to non-targeted messaging on landing pages. There is an opportunity to better align content with user intent and create optimized landing pages that drive conversions.</background>
<description>Design and develop targeted landing pages that align with specific content topics and optimize call-to-action (CTA) elements. Conduct usability tests with potential users to gather feedback on the design and content. Simulate the tracking of user interactions on these landing pages to create a comprehensive report on expected performance improvements.</description>
<constraints>Ensure that the landing page designs are consistent with PriGen's branding and adhere to best practices for SEO. Generate synthetic user interaction data to simulate different pathways through the landing pages, including bounce rates and conversion metrics. Maintain regulatory compliance for data collection methods implemented through these pages.</constraints>
</task>

<task>
<name>Launch Incentivized Referral Program</name>
<background>PriGen aims to leverage the existing user base through an incentivized referral program launching on January 15, 2024, to drive new user acquisitions with a target of 200 new sign-ups.</background>
<description>Finalize the integration of referral tracking systems and unique referral codes for users. Prepare promotional materials, including an email blast and social media graphics, to announce the program. Simulate user engagement scenarios including referral tracking and incentive redemption processes to ensure functionality before launch.</description>
<constraints>Ensure that user data handling complies with privacy regulations, particularly regarding referral credits and personal data usage. Generate synthetic user profiles and referral activity logs to test the system's functionality and robustness. Collect feedback from initial participants to improve the program post-launch.</constraints>
</task>
```

Figure 11: The generated tasks from *TaskWeave* using ChatGPT-4o-mini.

# I ITHC

## I.1 INTERNAL TEXT HIERARCHICAL CLASSIFICATION (ITHC)

Given a natural language input, the goal is to assign it a fine-grained internal category from a predefined three-level taxonomy and to generate a textual rationale for the prediction. Let $\mathcal{T}$ denote the space of input texts (e.g., paragraphs or documents), and $\mathcal{Y}$ the hierarchical label space, and $\mathcal{R}$ the space of natural language rationales that explain the assigned label. The task is to learn a function:

$$f_{\text{ITHC}}(T|\mathcal{D}_{\text{gen}}) = (y,\ r)\,, \quad T \in \mathcal{T},\ y \in \mathcal{Y},\ r \in \mathcal{R}$$

Where each label $y = (y^{(1)}, y^{(2)}, y^{(3)})$ corresponds to a structured taxonomy level: Category > Subcategory > Fine-Grained Label. The explanation $r$ describes why the label applies to the input. The pair $(y, r)$ is automatically derived from $T$ using a rule-based extractor applied to synthetic data $\mathcal{D}_{\text{gen}}$.

## I.2 CLASSIFICATION LABEL SYSTEM

To simulate enterprise dynamics in a realistic and controllable manner, it is crucial to ground agent behaviors in structured organizational knowledge. We propose a three-level hierarchical taxonomy $\mathcal{Y} = \{(y^{(1)}, y^{(2)}, y^{(3)})\}$ tailored for internal enterprise documentation. This taxonomy serves as the backbone for classifying documents, guiding agent planning, and aligning simulated workflows with real-world business structures.

Unlike general-purpose classification schemas, enterprise environments demand fine-grained labels that reflect actual operational divisions, team structures, and role-specific responsibilities. A single document may inform decisions at multiple organizational levels—ranging from strategic planning to task execution. Without a semantically grounded taxonomy, agentic systems struggle to perform accurate routing, delegation, or progress tracking.

Moreover, internal documents often follow recurring patterns in scope and structure. For example, a "User Engagement Report" typically belongs to the *Customer & Marketing* domain, under the *Customer Insights & Analytics* subcategory, and conveys user interaction data that guides acquisition or retention campaigns. A flat or ad-hoc labeling approach would obscure these functional signals and impair model interpretability.

The taxonomy is organized into three levels:

- **Category** ($y^{(1)}$): The top-level layer captures broad business areas such as *Human Resources* or *Operations*, aligning with major departments or role clusters in a company.

- **Subcategory** ($y^{(2)}$): The second level reflects domain-specific processes, e.g., *Employee Development & Training* or *Technology Research & Planning*, which typically correspond to team-level responsibilities or functional workflows.

- **Fine-Grained Label** ($y^{(3)}$): The bottom layer contains task- or report-level identifiers (e.g., `HR_TRAINING_FEEDBACK`), serving as atomic units of enterprise knowledge.

This structure mirrors the actual information granularity observed in internal communications and records. It supports flexible abstraction for both high-level planning and low-level execution. When combined with large-scale language models, such structured label spaces enable hierarchical reasoning, plan decomposition, and goal tracking in long-horizon simulations.

The taxonomy provides multiple downstream benefits:

- *Task Decomposition:* Enables agents to decompose goals into role-appropriate subgoals by leveraging label hierarchies.

- *Trace Explainability:* Enhances interpretability of agent decisions via label-aligned rationales.

- *Cross-role Collaboration:* Facilitates structured hand-offs between roles (e.g., from strategists to operators) by ensuring semantic continuity in document types.

- *Data Generation and Evaluation:* Supports automatic annotation and fine-grained evaluation in synthetic or semi-supervised corpora via label-conditioned rules.

A full listing of all categories, subcategories, and fine-grained labels is shown in Table 11. This taxonomy is not merely descriptive—it is designed as a functional schema to drive realistic simulations of enterprise workflows, planning structures, and role-based decision-making.

| Primary Category | Secondary Category | Tags (Tertiary Level) |
|---|---|---|
| Customer & Marketing | Campaigns & Promotions | MKT_CAMPAIGN_ANALYSIS
MKT_CAMPAIGN_PERFORMANCE
MKT_EVENT_OPERATION
MKT_PROMOTION_ANALYSIS |
| | Customer Insights & Analytics | CUST_ENGAGEMENT_REPORT
CUST_FEEDBACK_ANALYSIS
CUST_MARKET_ANALYSIS
MKT_DATA_REPORT |
| | User Growth & Profiling | MKT_USER_ACQUISITION
MKT_USER_CONVERSION
MKT_USER_PROFILE |
| Content & Media | Content Creation & Publications | CONTENT_BLOG_POST |
| Data & Technology Management | Data Quality & Infrastructure | DATA_COMPLETENESS_REPORT
DATA_QUALITY_ISSUE
OPS_SYSTEM_MONITORING |
| | Technology Research & Planning | OPS_NEW_TECH_RESEARCH
STRAT_TECH_INNOVATION_PLAN
STRAT_TECH_PARTNERSHIP_REPORT |
| Human Resources | Talent Acquisition & Onboarding | HR_NEW_EMPLOYEE_REPORT
HR_RECRUITMENT_PLAN
HR_RECRUITMENT_RECORDS |
| | Employee Development & Training | HR_TRAINING_FEEDBACK
HR_TRAINING_PROGRAM
HR_TRAINING_RECORDS |
| | Engagement & Compliance | HR_EMPLOYEE_ENGAGEMENT_REPORT
HR_EMPLOYEE_FEEDBACK
HR_COMPLIANCE_REPORT
HR_POLICY_DOCUMENT |
| Operations | Task Execution & Management | OPS_TASK_EXECUTION
TASK_EXECUTION_STATUS
TASK_EXECUTION_SUMMARY |
| Security & Compliance | Risk & Policy Management | SEC_COMPLIANCE_AUDIT
SEC_DATA_PROTECTION_GUIDELINE
SEC_POLICY_DOCUMENT
SEC_INCIDENT_RESPONSE_PLAN |
| Strategy & Innovation | Strategic Execution & Growth | STRAT_CROSS_DEPARTMENT_COLLAB
STRAT_IMPLEMENTATION_PLAN
STRAT_USER_EXPERIENCE_IMPROVEMENT
STRAT_USER_GROWTH_PLAN
STRAT_WEBINAR_IMPROVEMENT_STRATEGY |
| | Market & User Strategy | STRAT_MARKETING_STRATEGY
STRAT_USER_ENGAGEMENT_STRATEGY |
| User Experience & Research | Behavior & Feedback Analysis | UX_INTERACTION_ANALYSIS
UX_USER_BEHAVIOR
UX_USER_ENGAGEMENT
UX_USER_FEEDBACK_SUMMARY |
| | User Testing & Research | UX_USER_RESEARCH_REPORT
UX_USER_TESTING_REPORT |

Table 11: Internal Text Hierarchical Classification Label System

## I.3 EVALUATING MODEL OUTPUTS WITH ITHC

To systematically evaluate the consistency and semantic richness of document labels generated by different models, we propose a unified re-classification framework based on a three-stage self-consistency filtering pipeline using `GPT-4o-mini`. The full set of prompts used in this process is shown in Figure 12.

### Prompt for ITHC

```
"""
## AGENT ROLE
You are the Data Protection Officer (DPO) at PriGen, a major telecom enterprise. You are
responsible for classifying operational documents based on their content sensitivity,
ensuring compliance with privacy, security, and regulatory standards.Your primary
objective is to **analyze and categorize documents** into a **hierarchical classification
system**, marking those that contain **privacy-sensitive, legally protected, or
confidential business data**.The classification results are strictly in accordance with # #
EXPECTED OUTPUT EXAMPLE

## CLASSIFICATION SYSTEM
The classification follows a multi-level structure, starting with broad categories and
refining into more specific subcategories. Categories focus on **customer data,
employee records, financial transactions, telecom network information, security reports,
and legal compliance.**
{classificaction system}

## TASK INSTRUCTIONS
1. You will receive a document from telecom operations.
2. Analyze the document and classify it using the hierarchical classification system.
3. Assign the most specific applicable label(s) to the document.
4. If a document does not fit an existing label, propose a new label under the
appropriate category.
5. Output the classification result in **LIST** format.
6. Avoid adding extra commentary or explanation outside the final classification output.
7. Strictly output in the format of ##EXPECTED OUTPUT EXAMPLE
8. Can output multiple results

## EXPECTED OUTPUT EXAMPLE
("Category > Subcategory > Label", "reason")
Where:
-'"reason'"is a **short explanation** why the spansensitive and why this entity applies

## DOCUMENT TO BE CLASSIFIED:
{file_content}
"""
```

### Prompt for ITHC Audit

```
"""
## AGENT ROLE
You are a Classification Review Agent at PriGen, a major telecom enterprise. Your
responsibility is to **carefully review and validate existing classification labels**
assigned to telecom operational documents. Your goal is to ensure that each label is
**accurate, precise, and contextually appropriate**. You should only make changes to
labels that are **clearly incorrect, inconsistent, or misleading**. If a label is reasonable,
it should be retained to preserve labeling consistency. All changes must follow the
established hierarchical classification system.

## TASK INSTRUCTIONS
1. You will receive a document and its initial classification result.
2. Review whether each label is **broadly reflective** of the document content.
3. For each original label:
- If the label is **clearly wrong, misleading, or not related at all** to the document,
replace it with a more suitable one.
- If the label is **generally acceptable**, even if not perfect, **do not modify it**.
4. Be conservative in making changes. **Aim to minimize the number of revisions
unless strongly justified.**
5. Maintain the existing label hierarchy and structure.
6. Output the revised classification result in **LIST** format.
7. Do not include any extra comments outside the required format.
8. Be mindful not to overfit or over-classify — favor clarity and generality over detail.

## EXPECTED OUTPUT EXAMPLE
("Category > Subcategory > Label", "reason")
Where:
- "reason" is a **brief explanation** of why the document content fits this label.

## DOCUMENT TO BE REVIEWED:
{file_content}

## ORIGINAL LABELS:
{original_labels}
"""
```

Figure 12: The prompt of ITHC

In the first stage, `GPT-4o-mini` is prompted to assign a top-level category based solely on the original output from each base model. In the second stage, it performs a self-review by re-evaluating the same input document, but this time with access to its own first-stage prediction and reasoning process. In the third stage, it conducts another round of self-assessment, taking as input the output from the second stage.

This progressively reflective setup ensures that later evaluations are not independent reruns, but informed reassessments of previous reasoning. A label is considered stable and accepted only if it remains unchanged across all three stages.

This approach enables consistent and comparable analysis of outputs across six models: `GPT-4o-mini`, `Gemini-2.0-Flash`, `Deepseek-v3`, `Moonshot-v1-8K`, `LLaMA3.1-70B`, and `GLM-4-Flash`, regardless of the original document content.

As illustrated in Figure 13, the most frequent label across all models is `Customer & Marketing`. However, the semantic coverage varies considerably. `Gemini-2.0-Flash` exhibits the most balanced and comprehensive label distribution, covering all eight top-level categories—including low-frequency ones such as `Content & Media` and `Data & Technology Management`—demonstrating superior contextual sensitivity and label expressiveness.

By contrast, models like `Deepseek-v3` and `GLM-4-Flash` display a strong bias toward dominant categories, indicative of mode-seeking tendencies. `GPT-4o-mini` and `Moonshot-v1-8K` achieve broader coverage, though not to the extent of `Gemini-2.0-Flash`.

These results suggest that `Gemini-2.0-Flash` is particularly well-suited for classification tasks requiring fine-grained label diversity and robust representation of long-tail categories.

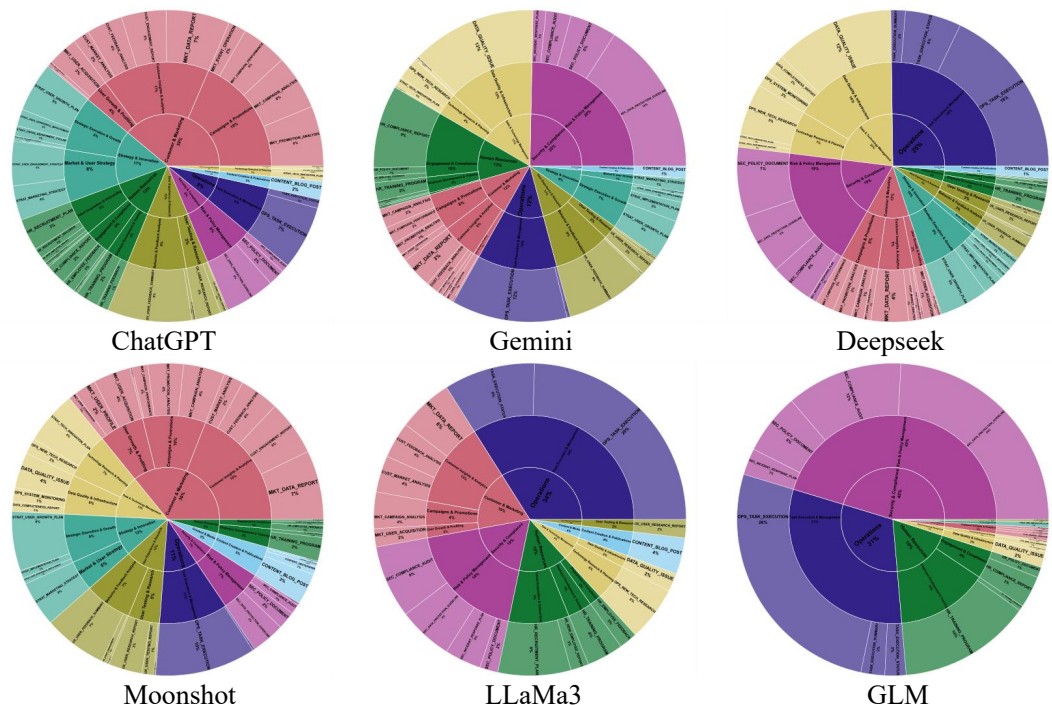

Figure 13: ITHC outputs of six models

## J    MAS CHECKPOINT

This section provides an in-depth description of the MAS Checkpoint mechanism, which serves as the backbone for task tracking and completion verification in our system. The methodology simulates enterprise-grade planning and execution cycles, leveraging both language models and structured sliding window logic to achieve robust evaluation.

### J.1    TASK POOL CONSTRUCTION

For each evaluation week $t$, we define a unified task pool $\mathcal{T}_t$ composed of newly planned tasks and previously uncompleted tasks from a retrospective window of $w$ weeks:

$$\mathcal{T}_t = T_t \cup \bigcup_{i=1}^{w} T_{t-i}^{(u)} \tag{8}$$

**Where:**

- $T_t$ is the set of newly generated tasks for the current evaluation week $t$.

- $T_{t-i}^{(u)}$ denotes the set of uncompleted tasks from week $t - i$.

- $w$ is the length of the sliding window, empirically set to 4 to match common enterprise monthly cycles.

### J.2    LLM-BASED TASK DECOMPOSITION

To convert free-form planning text into structured task lists, we prompt a language model to act as a task planning expert. The output is constrained to be a Python list of clearly defined, actionable items. We avoid explicitly requesting a fixed number of tasks; instead, we guide the model through prompt structure, examples, and abstraction control.

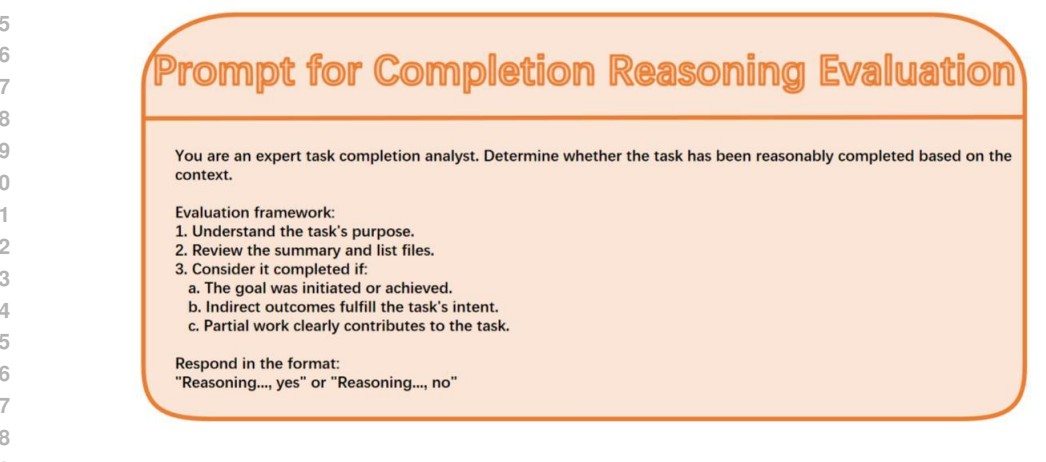

Figure 14: Prompt used for Task Decomposition

**Prompt Template for Task Decomposition**

### J.3 DUAL-MODEL TASK COMPLETION EVALUATION

Each task $\tau \in \mathcal{T}_t$ is evaluated independently by two large language models: Gemini 2.5 Flash and GPT-4o-mini. These models determine whether the task is considered `Completed` or `Uncompleted` based on current documentation $C_t$.

$$f_{\text{LLM}}(\tau, C_t) \rightarrow \{\text{Completed}, \text{Uncompleted}\} \tag{9}$$

Where $f_{\text{LLM}}$ is the evaluation function powered by either language model. $\tau$ is a single task to evaluate. $C_t$ represents the execution records of week $t$, including summaries and structured lists.

### J.4 PROMPT TEMPLATE FOR COMPLETION EVALUATION

Figure 15: Prompt used for Completion Reasoning Evaluation

### J.5 SLIDING WINDOW LIFECYCLE TRACKING

If a task remains uncompleted, it is carried into the next week's evaluation pool, provided it falls within the window constraint:

$$\tau \in T_t^{(u)} \Rightarrow \tau \in \mathcal{T}_{t+1} \quad \text{iff} \quad t - \texttt{created\_week}(\tau) < w \tag{10}$$

Where $\texttt{created\_week}(\tau)$ denotes the week when task $\tau$ was originally created.

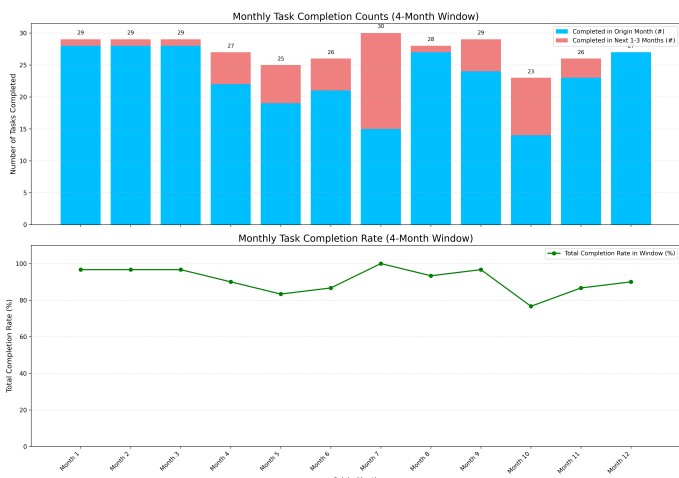

Figure 16: Monthly completion counts and rates within a 4-month window.

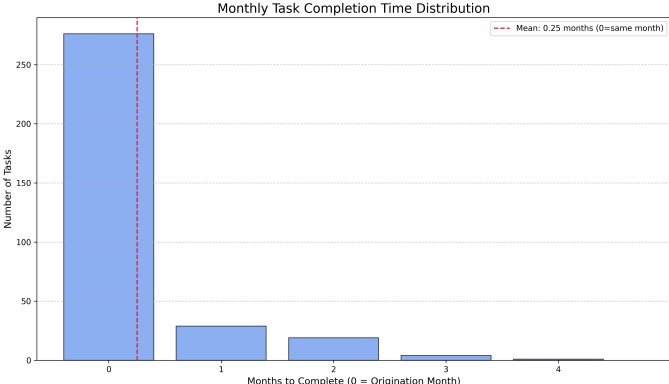

Figure 17: Monthly completion time distribution.

As shown in Figure 17, over 85% of tasks are completed within the same month, indicating the effectiveness of our LLM-generated decomposition in producing tractable, scoped workloads.

Figure 16 further confirms that tasks not completed in the origin month are typically recovered in subsequent months (within a 4-month window), supporting the robustness of our window-based carry-over mechanism.

Finally, To validate the reliability of LLM judgments, 10% of evaluated tasks are randomly sampled and reviewed manually. If the disagreement exceeds 10%, full re-evaluation is conducted. This feedback loop also produces high-quality supervision data for potential model fine-tuning.

### J.6 ABLATION STUDY AND COMPLETION RATE ANALYSIS

To evaluate the role of documentation $C_t$ in supporting model judgment, we conduct an ablation experiment. We define the original completion function as: $\hat{y}_{\text{full}}(\tau) = f_{\text{LLM}}(\tau, C_t)$ Then we define a reduced variant: $\hat{y}_{\text{ablated}}(\tau) = f_{\text{LLM}}(\tau, C_t')$ with $C_t' \subset C_t$ The empirical drop in completion rate

is computed as: $\Delta R = R_{\text{full}} - R_{\text{ablated}} \approx 50\%$ Where: $\hat{y}$ is the predicted completion decision. $R$ denotes the proportion of tasks labeled `Completed`. $C'_t$ excludes critical context such as status summaries or key execution lists. This significant drop in performance underscores the necessity of comprehensive context and validates our system's sensitivity to semantically grounded documentation. It also serves as an implicit test for whether the model truly understands the evidence rather than relying on shallow keyword features.

## K  OSSD

Using LLMs to annotate or verify LLM outputs has proven effective in prior work across instruction generation (Wang et al., 2023), plan synthesis (Liu et al., 2024b), and trajectory validation (Gao et al., 2025). Building on these insights, we extend this paradigm to privacy annotation in simulated enterprise settings. We design a multi-stage structured prompting strategy for identifying and validating privacy-sensitive text spans, termed **OSSD (Organizational Sensitive Span Detection)**. This mechanism simulates the role of a Data Protection Officer (DPO) and enables scalable, structured supervision for downstream tasks OSSD.

Each annotation process consists of the following two stages:
**Label Stage:** Given a task, paragraph, or document, the system identifies privacy-relevant content. For OSSD, this involves extracting sensitive spans and assigning privacy category labels with accompanying textual explanations. For ITHC, the system predicts a hierarchical label and provides a rationale for the classification.
**Check Stage:** A validation module re-evaluates the predicted label and explanation for consistency, specificity, and semantic adequacy. This step mimics the human review process, correcting errors such as generic reasoning or misclassified categories.

### K.1  Annotation Pipeline Overview

The OSSD procedure consists of three stages: broad discovery, contextual refinement, and reasoning-based validation. **Stage 1: Broad Discovery** The model is prompted to extract a wide range of potentially sensitive spans. The prompt prioritizes recall and avoids committing to specific type assignments. The prompt template is:

Figure 18: Prompt used in Stage 1: Broad Discovery

The model returns:
$$\mathcal{B}_x = \{(e_i, \text{``UNCERTAIN''})\}_{i=1}^{k} \tag{11}$$
**Where:** $e_i$ is a candidate privacy-relevant span.

**Stage 2: Contextual Refinement** In this stage, the system is prompted to re-evaluate each span from Stage 1 and assign a type $t_i \in \mathcal{T}$ from a predefined taxonomy, with a natural language explanation $r_i$. The refined annotation set is:
$$\mathcal{S}_x = \{(r_i, e_i, t_i)\}_{i=1}^{n} \tag{12}$$

Where: $e_i$: the refined entity. $t_i$: the specific privacy type from the taxonomy. $r_i$: the reasoning for labeling.

The prompt is:

Figure 19: Prompt used in Stage 2: Contextual Refinement

**Stage 3: Reasoning-Based Validation** This phase performs critical audit of Stage 2 outputs. The goal is to revise or discard weak, vague, or inconsistent annotations. The final output is:

$$\mathcal{S}'_x = \texttt{Validate}(\mathcal{S}_x) \tag{13}$$

The validation prompt asks the model to simulate logical DPO review:

Figure 20: Prompt used in Stage 3: Reasoning-Based Validation

### K.2 PROMPT ENGINEERING AND STABILITY

To ensure robustness, we instantiate all stages with the same model (GPT-4o-mini), but vary the temperature across stages to simulate cognitive diversity:

- Stage 1: Temperature $T = 0.8$ (encouraging exploration).
- Stage 2: Temperature $T = 0.5$ (balanced contextual judgment).
- Stage 3: Temperature $T = 0.3$ (focused logical validation).

This setup stabilizes the pipeline by introducing soft redundancy. We further evaluate consistency by computing the Jaccard similarity:

$$\text{Jaccard}(\mathcal{S}_x, \mathcal{S}'_x) = \frac{|\mathcal{S}_x \cap \mathcal{S}'_x|}{|\mathcal{S}_x \cup \mathcal{S}'_x|} \tag{14}$$

Where $\mathcal{S}_x$: refined annotations from Stage 2 $\mathcal{S}'_x$: validated annotations from Stage 3. A match is defined on both $(e_i, t_i)$ pairs.

### K.3 ROLE SIMULATION AS DPO

This three-stage structure mirrors real-world workflows of a Data Protection Officer. In actual enterprise settings, the DPO first flags potentially sensitive content (analogous to Stage 1), then audits those decisions for policy compliance and annotation quality (Stage 2), and finally performs logical consistency checks (Stage 3).

By embedding explanatory rationales $r_i$ and enabling feedback-driven correction, the OSSD process supports transparency and interpretability. With such two core principles in high-stakes governance environments. Beyond this, our enterprise-level annotation setup demonstrates the model's ability to surface multiple categories of sensitive internal information, such as strategic plans, employee identifiers, and operational intents. This capability not only verifies that OSSD effectively simulates the real-world internal workflows of data protection within enterprises, but also confirms its potential as a practical tool for constructing large-scale privacy datasets. These annotated internal spans serve as high-quality supervision signals $\{(x, r_i, e_i, t_i)\}$, which can be used to train models for downstream privacy tasks and facilitate generalization of privacy reasoning across domains.

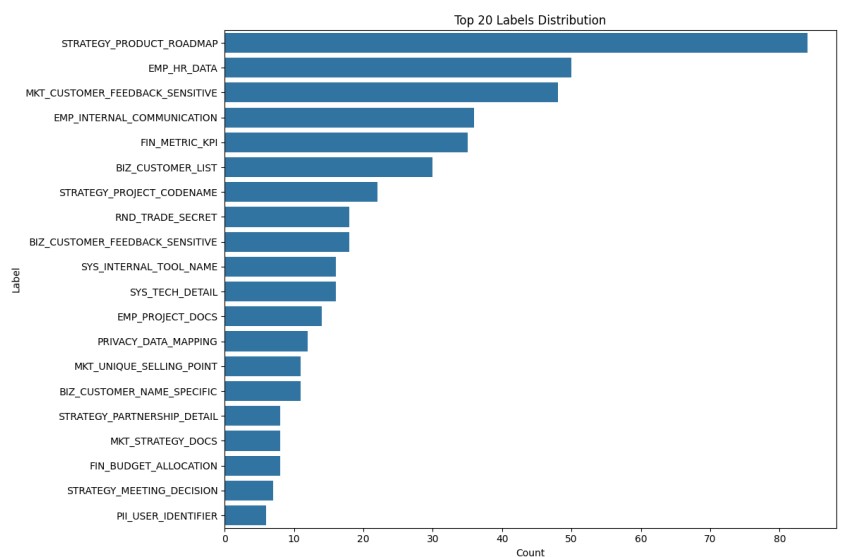

Figure 21: Label distribution in our system.

## L GENERALZABILITY

### L.1 FINANCIAL COMPANY

We evaluate *TaskWeave* by simulating the month-long operation of a representative financial company, denoted as Fin. The organization is composed of 15 role-specialized agents distributed across 3 organizational tiers (Tier 1–3).

**Tier 1.** Chief Executive Officer

**Tier 2.** Chief Investment Officer, Chief Risk Officer, Chief Operations Officer

**Tier 3.** Equity Trader, Fixed Income Analyst, Portfolio Assistant, Credit Risk Analyst, Market Risk Specialist, Internal Auditor, Settlement Officer, Compliance Associate, Fund Accountant,Relationship Manager, Client Onboarding Specialist

## L.2 MANUFACTURING COMPANY

We evaluate *TaskWeave* by simulating the month-long operation of a representative automotive manufacturing company, denoted as Manu. The organization is composed of more than 30 role-specialized agents distributed across 4 organizational tiers (Tier 1–4).

**Tier 1.** Chief Executive Officer

**Tier 2.** Chief Product Officer, Chief Marketing Officer, Chief Operations Officer, Chief Technology Officer

**Tier 3.** Production Manager, Quality Manager, Maintenance Manager, Logistics Manager, Digital Systems Manager

**Tier 4. Under Production Manager:** CNC Operator, Tool Setter, Assembly Worker, Soldering Technician, Robotics Technician

**Under Quality Manager:** Final Inspector, Visual Inspector, Compliance Auditor

**Under Maintenance Manager:** PLC Technician, Wiring Technician, Machine Repairman, Hydraulics Technician, Predictive Maintenance Engineer

**Under Logistics Manager:** Receiving Clerk, Forklift Operator, Shipping Coordinator, Boxing Technician, Packaging Design Engineer

**Under Digital Systems Manager:** MES Analyst, Data Security Officer, AI Vision Specialist, Traceability Systems Specialist

## L.3 GOVERNMENT AGENCY (GOV)

We evaluate *TaskWeave* by simulating the month-long operation of a representative government agency, denoted as Gov. The organization is composed of around 100 role-specialized agents distributed across 5 organizational tiers (Tier 1–5).

**Tier 1.** Minister

**Tier 2.** Director of Strategic Planning, Director of Public Infrastructure, Director of Social Programs, Director of Policy Analysis, Director of Inter-Agency Coordination

**Tier 3.** Chief of Strategic Planning, Chief of Infrastructure Development, Chief of Social Programs, Chief Policy Analyst, Chief Administrative Officer

**Tier 4. Under Chief of Strategic Planning:** Senior Planning Manager, Urban Strategy Manager

**Under Chief of Infrastructure Development:** Infrastructure Project Manager, Public Health Program Manager

**Under Chief of Social Programs:** Social Policy Manager, Sustainability & Environment Manager

**Under Chief Policy Analyst:** Policy Research Supervisor, Regional Coordination Manager

**Under Chief Administrative Officer:** HR & Admin Manager, Budget & Resource Allocation Manager

**Tier 5. Under Senior Planning Manager:** Planning Officer, Municipal Data Analyst, Health Policy Assistant, Citizen Complaint Registrar

**Under Urban Strategy Manager:** Administrative Clerk, Digital Records Clerk, Public Service Trainee

**Under Infrastructure Project Manager:** Construction Analyst, Infrastructure Surveyor, Strategic Affairs Associate

**Under Public Health Program Manager:** Stakeholder Liaison, Legal Compliance Officer, Public Opinion Analyst

**Under Social Policy Manager:** Welfare Program Officer, Community Outreach Assistant, Procurement Assistant

**Under Sustainability & Environment Manager:** Citizen Engagement Officer, Smart City Systems Technician, Site Inspector

**Under Policy Research Supervisor:** Field Research Associate, Education Policy Aide, Audit Support Officer

**Under Regional Coordination Manager:** Regional Coordination Manager, Training Coordinator, Statistical Reporting Assistant

**Under HR & Admin Manager:** HR Specialist, Transportation Planner, Scheduling & Logistics Clerk

**Under Budget & Resource Allocation Manager:** Project Evaluation Assistant, E-Government Support Officer, Facility Oversight Technician

## M  KEY TASKS GENERATED BY TASKWEAVE

To understand the structural importance of tasks within the company's yearly operations, we analyze the directed task dependency graph generated by the multi-agent simulation. In this graph, nodes represent atomic business tasks and edges indicate task-level dependencies or references.

The in-degree of a task indicates how many other tasks rely on it. A higher in-degree implies broader influence and strategic importance. We present the Table 12, reflecting their centrality in enabling or being referenced by other activities.

Table 12: Top 20 Tasks by In-Degree in the Task Dependency Graph

| Task Name | In-Degree |
| --- | --- |
| Conduct_Keyword_Research_for_SEM_Campaign | 103 |
| Monitor_and_Analyze_Referral_Program_Performance | 98 |
| Analyze_Performance_of_Referral_Program_After_Launch | 88 |
| Optimize_Email_Marketing_Campaigns | 75 |
| Launch_PriGen_Referral_Rewards_Program | 58 |
| Create_Focus_Groups_for_UI_Improvement_Feedback | 58 |
| Enhance_Incentivized_Referral_Program | 53 |
| Launch_Incentivized_Referral_Program | 52 |
| Enhance_Social_Media_Advertising_Campaign | 51 |
| Refine_SEM_Targeting_Strategy | 50 |
| Revise_UI_Based_on_Focus_Group_Feedback | 50 |
| Formalize_Strategic_Partnership_Agreements | 48 |
| Monitor_and_Optimize_Digital_Marketing_Campaigns | 45 |
| Create_Distinct_SEM_Campaign_Structures | 45 |
| Design_Ad_Copy_for_SEM_Campaigns | 42 |
| Prepare_Co-Marketing_Materials_with_Strategic_Partners | 36 |
| Collect_and_Analyze_User_Feedback_on_New_Features | 33 |
| Enhance_the_Incentivized_Referral_Program | 32 |
| Refine_SEM_Targeting_Strategy_for_Improved_Lead_Generation | 32 |

To complement the in-degree analysis, we now examine tasks with the highest out-degree, as summarized in Table 13. Out-degree reflects how many downstream tasks each task depends on or triggers. Tasks with high out-degree typically represent coordination hubs, campaign rollouts, or integrative planning activities, indicating their central role in initiating or orchestrating complex workflows.

Table 13: Top 20 Tasks by Out-Degree in the Task Dependency Graph

| Task Name | Out-Degree |
|---|---|
| Review_and_Optimize_User_Engagement_Strategies_based_on_Feedback | 59 |
| Monitor_and_Optimize_Digital_Marketing_Campaigns | 24 |
| Engage_in_Social_Media_Campaign_Analysis | 24 |
| Optimize_Digital_Marketing_Campaigns_Based_on_Performance_Data | 24 |
| Conduct_Deep_Dive_Analysis_of_SEM_Performance_Metrics | 22 |
| Initiate_Keyword_Optimization_for_SEM_Campaigns | 21 |
| Revise_and_Enhance_the_Onboarding_Feedback_Mechanism | 21 |
| Enhance_User_Feedback_Mechanism_through_Incentives | 20 |
| Initiate_Content_Marketing_Strategy | 19 |
| Optimize_User_Feedback_Collection_Post-Onboarding | 19 |
| Implement_Feedback_Mechanisms_for_Onboarding_Reviews | 18 |
| Launch_Interactive_Social_Media_Campaigns | 18 |
| Monitor_and_Optimize_SEM_Campaign_Performance | 17 |
| Review_and_Finalize_A/B_Testing_Strategies_Across_Ad_Campaigns | 17 |
| Launch_Content_Marketing_Initiatives_with_Focused_Webinars | 16 |
| Implement_Lookalike_Audiences_for_Marketing_Campaigns | 16 |
| Conduct_A/B_Testing_for_SEM_Ad_Copies | 16 |
| Finalize_A/B_Testing_Adjustments_on_Ad_Copies | 15 |
| Implement_A/B_Testing_for_Ad_Copies | 15 |
| Execute_Targeted_Social_Media_Engagement_Campaigns | 15 |

# N   INTERACTION WITH THE EXTERNAL ENVIRONMENT

Our work focuses on designing a more realistic and generalizable multi-agent system that simulates enterprise operations.

External-environment modeling is important and complex: different enterprises face different environments, the required granularity must be designed case by case, and different simulation needs lead to different external-environment design requirements. This paper concentrates on high-quality and generalizable modeling of the *intra-enterprise* scenario.

To verify that TaskWeave can effectively couple with external environments, we supplement one experiment that demonstrates the framework's bidirectional interaction capability:

- **ENV→MAS:** TaskWeave receives external information via external-event injection.

- **MAS→ENV:** Agents are configured with tools to act upon the environment.

## N.1   EXTERNAL-EVENT INJECTION

To verify that external events can influence TaskWeave, we inject realistic, real-world-based incidents and subsequently assess agent awareness through structured interviews; key words are extracted from the newly generated planning documents to quantify the resulting strategic shift.

### N.1.1   EXPERIMENTAL CONFIGURATION

The experiments are based on the CompanyA scenario, using 4o-mini as the backbone model. The simulation spans one quarter, with events injected during the quarterly planning stage. We collected 70 events in total across the three domains of Policy, Economic, and Technology, of which 15 were injected into the experiments. Several representative events are shown in Table 14.

Table 14: Representative Injected Events

| Title | Catalog | Real World Basis |
|---|---|---|
| Digital Resilience Certification Mandate | Policy | EU CRA political agreement reached 30 Nov 2023; certification bodies already booking audits for 2025. |
| Regulatory Tightening | Policy | During the first week of PIPL in Aug 2021, dozens of SaaS firms received urgent self-inspection notices. |
| Cross-Border Data Ban | Policy | Cyber-Security Law 2017 forced Apple iCloud China migration. |
| Macro Downturn | Economic | Q2 2022 China GDP printed 0.4%; multiple SaaS earnings calls cited budget freezes. |
| Interest-Rate Spike | Economic | Fed's first 25 bp hike in Mar 2022 triggered a global SaaS de-rating. |
| CPI 8% Inflation Shock | Economic | U.S. Bureau of Labor Statistics, 10 Jun 2022. |
| WASM-First Web Frameworks Reshape Client-Side Performance Baselines | Technology | Based on trends in Qwik, WASM adoption in Figma-like editors, and Cloudflare Workers + UI thread offload frameworks (2024–2025). |
| Zero-Day Cascade | Technology | Dec 2021 Log4j crisis required 48-hour global patch cycle. |
| LLM-Driven Collaboration Norm Shift | Technology | Trend based on Mixtral, Claude 3, DBRX, and OpenDevin integration patterns in 2024–2025. |

### N.1.2 EVENT IMPACT

**Response Strategy**    We extract the key words in the plan as follows:

- **Original:** search engine marketing, tiered support system, user acquisition strategies, predictive analytics, AI-driven features.

- **Policy:** lifecycle assessments, Corporate Sustainability Reporting Directive, Cross-Border Data Ban, stakeholder engagement, resource allocation challenges, compliance documentation.

- **Economic:** zero-based budgeting, operational cash flow management, performance metrics, user acquisition strategies, churn risk mitigation, consultative selling techniques.

- **Technology:** digital marketing initiatives, document loading optimization, AI feature development, operational robustness, zero-trust architecture.

Agents also exhibit situational awareness (affirmative interviews) and generate corresponding strategic responses, indicating that TaskWeave supports adaptive, context-aware planning in response to external dynamics

In conclusion, TaskWeave demonstrates sensitivity to external environmental factors.

### N.2 TOOL INVOCATION

In TaskWeave, every agent can be initialized with a set of tools, interact with real-world products and operate with the local operating system. These tools enhance the MAS's ability to interact with the external environment, enabling the system to exert real-world effects and yielding more authentic task outcomes. To demonstrate that TaskWeave can indeed engage with the outside world, we conducted the following supplementary experiment.

### N.2.1 EXPERIMENT CONFIGURATION

The experiment adopts the CompanyA architecture; we equip selected agents with 63 tools spanning three categories (SQL, Social-Media, and Office) and run 30 distinct tasks. The detailed tool assignment for each agent is listed in Table 15.

Table 15: Agent–Tool Assignment in CompanyA (✓ = equipped)

| Agent | SQLite | MotherDuck | Word | Excel | Email | Twitter |
|---|---|---|---|---|---|---|
| Backend Engineer | ✓ | ✓ | | | | |
| Data Analyst | | ✓ | | ✓ | | |
| Product Manager | | | ✓ | ✓ | ✓ | |
| Marketing Specialist | | | | ✓ | ✓ | ✓ |
| Customer Success Manager | | | ✓ | | ✓ | ✓ |
| DevOps Engineer | | ✓ | | | ✓ | |
| HR Manager | | | ✓ | | ✓ | |
| Technical Support Engineer | | | | | | |
| UI/UX Designer | | | | | | |
| Frontend Engineer | | | | | | |
| QA Engineer | | | | | | |

### N.2.2 TOOLS IMPACT

**Impact by Tool Category**

**SQL** When data-analysis agents invoked SQL tools, they retrieved critical information from the database. The database acted as a centralised shared memory for the MAS, enhancing collaborative decision-making. During the experiment, **33 SELECT queries** and 8 additional SQL operations were executed.

**Social-Media** When agents responsible for market promotion triggered social-media tools, they proactively posted tweets and sent e-mails to partners. Company-wide announcements were also distributed via e-mail, reproducing internal communication workflows. Throughout the experiment, the MAS published **8 promotional tweets** and sent **46 e-mails**.

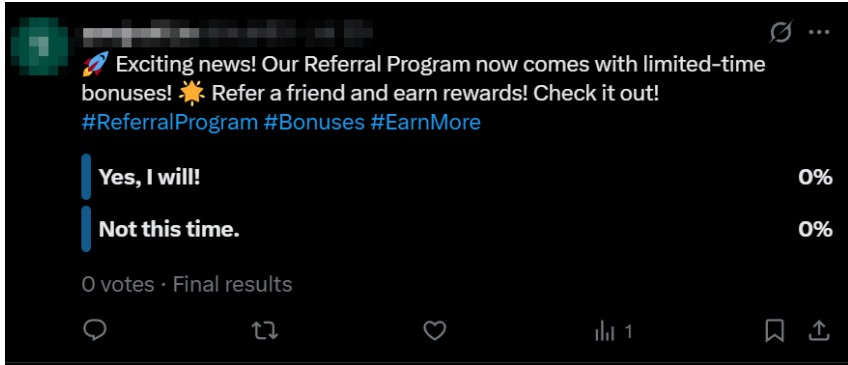

Figure 22: Example tweet sent by an agent via the social-media tool.

**Unlock Amazing Bonuses with Our Referral Program!**

From:

To:nancymorrison <nancymorrison@prigen.com>

Subject: Unlock Amazing Bonuses with Our Referral Program! Dear Valued User, We're thrilled to announce a ** When you refer friends to join us, both you and your friend can unlock exclusive bonuses! This is a fantasti Time Bonuses*: Act now to earn more rewards for each successful referral! 💬 *Hear from Our Happy Users*: Jamie L. ✨ *Seamless Process*: Referring friends is simple. Just share your unique referral link and watch the

Figure 23: Example email sent by an agent via the social-media tool.

**Office** When an agent activated office-suite tools, it generated Word and Excel reports. These auditable documents persist in the external environment, providing the MAS with reusable data assets. During the experiment, the MAS produced **51 office documents**.

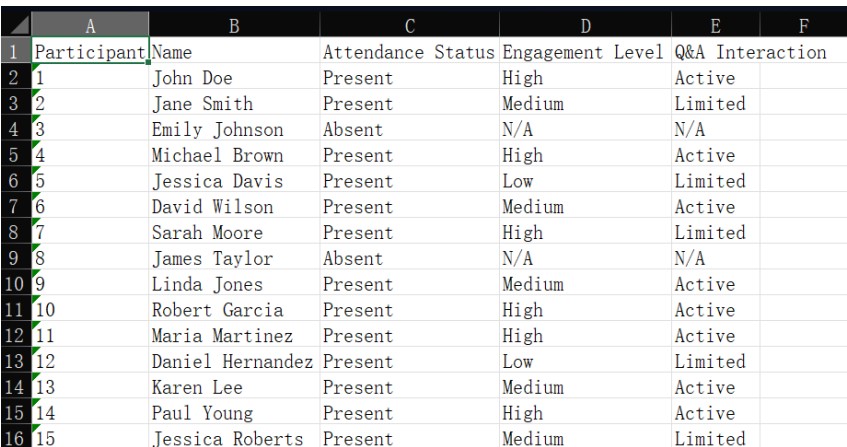

Figure 24: Example excel created by an agent via the office tool.

In summary, agents in TaskWeave can influence the external environment through tool usage.

