# OpenReview forum: "Multi-agent Imitates Enterprise Dynamics"
_ICLR.cc/2026/Conference — ICLR 2026 Conference Withdrawn Submission_

### Official Review · Reviewer_CfVf · 2025-10-26

**Soundness:** 2
**Presentation:** 2
**Contribution:** 2
**Rating:** 2
**Confidence:** 3

**Summary:**

This paper introduces TaskWeave, a multi-agent framework designed to simulate enterprise dynamics by orchestrating LLM-based agents across three interconnected layers: strategic planning, tactical delegation, and operational execution. Specifically, at the strategic level, the system uses the FPDA cycle to perceive internal and external conditions and refine global intent. At the tactical level, agents switch between orchestrating and executing roles to break down goals, delegate tasks, and align efforts. At the operational level, agents use structured memory and tools to access historical data, resolve task dependencies, and support context-based reasoning and traceable outputs.

**Strengths:**

1. This paper addresses an important issue by focusing on a specific and complex application scenario: enterprise dynamics. Unlike previous multi-agent systems that were more generalized and not tailored to a particular context, this work zeroes in on the intricacies of enterprise operations.

2. This paper effectively simulates the complexity of enterprise dynamics through a three-level orchestration, including strategic planning, tactical delegation, and operational activities. Such design is attractive.

**Weaknesses:**

1. This paper's main issue is its confusing and disorganized writing, which makes many parts difficult to understand. The use of complex and unclear terminology to explain variables like "structured task bundle" significantly hinders readability. This confusion begins with Equation (4), where the meaning of the subscript $i$ and the definitions of functions $\mathcal{C}$ and $\mathcal{V}$ are not clearly explained. Examples would be helpful to clarify these concepts.

2. Additionally, the paper lacks explanations for certain variables. For instance, $\mathcal{N}_{sibling}^+$ on line 221 and $\mathbb{H}$ on line 257 are not adequately defined. The order of presentation is also problematic; for example, the symbols $\delta$ and $\xi$ introduced on line 245 are only explained much later, creating significant reading difficulties.

3. The paper also contains several confusing typos. For example, there are two Equation (6)s, which is a clear mistake.

**Questions:**

See Weakness.

---

### Official Review · Reviewer_6cxA · 2025-10-28

**Soundness:** 1
**Presentation:** 2
**Contribution:** 2
**Rating:** 2
**Confidence:** 3

**Summary:**

This work proposes TaskWeave, a multi-agent LLM-driven framework that imitates complex enterprise dynamics and operates at three levels: strategic planning, tactical delegation, and operational activities. TaskWeave is evaluated through a year-long simulation of an example SaaS company, where the main judgment is conducted via LLM-as-a-Judge and human experts. The proposed framework outperforms the single-agent baseline over six LLM backbones. In addition, TaskWeave is further extended to Fin, Manu, and Gov domains for studying the generalization of the proposed framework. TaskWeave serves as a foundation for synthesizing enterprise data.

**Strengths:**

1. TaskWeave presents as a novel multi-agent framework for imitating enterprise dynamics, which can be valuable for generating enterprise-related data, helping real-world enterprise decision making, and breeding startup companies.

2. The three-level orchestration that includes strategic, tactical, and operational strategies for multi-agent coordination is insightful and a suitable structure for modelling the enterprise behavior.

3. The experiments are conducted over a range of state-of-the-art LLM backbones, where TaskWeave outperforms a single-agent baseline substantially in both LLM score and Human evaluations. TaskWeave also yields richer spans with lower API costs in the OSSD task.

**Weaknesses:**

1. Though TaskWeave imitates the enterprise dynamics, the underlying multi-agent framework involves heavy human engineering and prompt designs for characterizing the role of each agent. The high dependence of human prompt engineering and the lack of automation for imitating enterprise dynamics can strongly limit the generalization of the framework for customization.

2. In Table 1, the evaluation is conducted over a single-agent baseline. However, there are many prior LLM-based MAS frameworks (as listed in Table 6, e.g. AutoGen, MetaGPT) that can be customized into enterprise imitation, which was ignored in the main experiments. In addition, the human evaluation is conducted over a very small group of people with only 4 domain experts, which may expose bias in the evaluation. Some metrics are ill-defined: for instance, the 4:4:2 distribution between Tech, Mkt, Strat is overfitted to a special style of enterprise structure, which makes the KL divergence study misleading.

3.  To the best of my understanding, this framework will be useful when it is evaluated against a real-world enterprise example with the history of the growth and the internal decision-making. However, the main study conducted in this work only involves a year-long simulation of a single company without grounding with real-world dynamics.

4. Though the author states ablation analysis in Appendix E, the influence of each module in the 3-level orchestration is not quantitatively ablated.

In line 352, there is a missing Appendix reference that can be fixed.

**Questions:**

1. To what extent is the proposed framework automated? Does the agent profile require human engineering or automatically generated by only conditioning on the background information?

2. In Eq 6, how is the task assigned to each agent precisely? How is the alignment computed?

**Details Of Ethics Concerns:**

This work involves the generation of enterprise data that may expose sensitive human information.

---

### Official Review · Reviewer_YZQq · 2025-10-30

**Soundness:** 2
**Presentation:** 2
**Contribution:** 2
**Rating:** 4
**Confidence:** 3

**Summary:**

The paper proposes TaskWeave, a llm based multi-agent framework for imitating complex enterprise dynamics. The system operates at three levels - strategic, tactical and operational. The authors evaluate TaskWeave through a year-long SaaS company simulation, comparing it against single-agent baselines across multiple dimensions including task generation quality, role assignment, plan propagation, and task execution. Additional experiments demonstrate generalizability across financial, manufacturing, and government domains.

**Strengths:**

1. The authors tackle an interesting and difficult problem which is simulating enterprise dynamics.
2. The authors evaluate the method across multiple domains to showcase its generalisability
3. Simulation along four level temporal hierarchy to enable long horizon planning is something absent in existing multi-agent evaluation setups.

**Weaknesses:**

1. Novelty - The paper lacks significant novelty and mostly becomes an engineering work. The paper says it introduces FDPA, however most of its components correspond to popular and well researched techniques in agentic systems( Formulate -> Task Decomposition, Partition -> Planning/Task Generation, Diagnose -> Aggregation and Summarisation)
2. Method section is unclear and difficult to follow. The method section includes several symbols and terms which are not well defined. Many components are defined at a high level without implementable details. For example
      - How does the decompose operator D (eq 5) work?,
      - What constitutes a dependency query?,
      - how does $\phi$ score alignment (eq 6)?
3. Evaluation - While some of the evaluation metrics proposed are interesting, it lacks important enterprise metrics such as cost, ROI, success metrics, execution efficiency etc. Some of the evaluation details are also unclear such as how task execution is evaluated. The authors also skip evaluation against existing multi-agent framework such as MetaGPT.
4. Missing Details - The paper does not talk about cost and resource requirements for the framework. It also skips significant implementation details such as the prompts for each of the 14 role-specialized agents as well as prompts for llm-as-a-judge evaluation.

**Questions:**

1. How are individual agents’ outputs judged for correctness? How are the sub-tasks evaluated for completion?
2. How does the framework handle scenarios where certain agents fail to accomplish their tasks?
3. What are the computational and token costs and how does it compare with other multi-agent methods?

Suggestions:
1. Improve the clarity of the methods section by providing a running example to help understand the various components. Consider reducing mathematical notations that does not aid in understanding. Replace high-level symbolic descriptions with implementable algorithms or detailed prompt recipes.
2. Compare against multi-agent baselines. Include enterprise relevant metrics.

Minor Comments:
1. Line 73 - ...  generate interdependent tasks -> incorrect grammar
2. Line 352 - appendix reference is missing.

---

### Official Review · Reviewer_SrAv · 2025-11-01

**Soundness:** 3
**Presentation:** 3
**Contribution:** 2
**Rating:** 6
**Confidence:** 3

**Summary:**

This paper introduce TaskWeave, an LLM-based multi-agent framework for simulating complex enterprise dynamics. The system operates at three hierarchical levels: 1) Strategic level. Uses a Formulate-Partition-Diagnose-Align control cycle for long-horizon planning.
2) Tactical level. Agents alternate between orchestrator and executor roles for task delegation. 3) Operational level. Context-aware execution with memory, dependency tracking, and tool access. The authors evaluate TaskWeave through year-long simulations of SaaS organizations, measuring task generation quality, role assignment distribution, plan completion rates.

**Strengths:**

1. The motivation is clear and interesting. The three-level architecture grounded in control theory, strategic management, and division-of-labor theory. Also, the adaptation of traditional control loops (PDCA) to multi-agent hierarchical planning is interesting and potentially generalizable.

2. TaskWeave generates usable synthetic data (e.g., enterprise documents, privacy annotations, hierarchical labels) that can support compliance and enterprise analytics research. The OSSD and ITHC tasks demonstrate meaningful downstream value.

3. Evaluation across six different LLM backbones provides comprehensive evidence.

4. The writing is clear and easy to follow.

**Weaknesses:**

1. I have some confusion with the Arbitrary Ground Truth. For example, "Ideal" task distribution (40% Tech, 40% Marketing, 20% Strategy). Why is this correct?

2. The framework design is multi-agent, which involves the closed models and tool-calling. It may consume a significant amount of tokens. What are the computational costs (tokens, time, money) for year-long simulations?

3. In the framework, several steps rely on the Prompt setup. How sensitive is the system to organizational structure, role definitions, and prompt design?

**Questions:**

See in Weaknesses.

---

### Note · Authors · 2025-11-19

**Comment:**

After carefully reviewing the reviewers’ comments, we believe that the current version of the paper requires  revision and further experiment. As these revisions cannot be completed within the rebuttal schedule, we would like to request the withdrawal of the submission.

**Withdrawal Confirmation:**

I have read and agree with the venue's withdrawal policy on behalf of myself and my co-authors.